# Distributed multi-parameter quantum metrology with a superconducting quantum network

Jiajian Zhang[1,5], Lingna Wang [2,5], Yong-Ju Hai [1,5], Jiawei Zhang[1,3,5], Ji Chu [1], Ji Jiang[1], Wenhui Huang[1], Yongqi Liang[1,3], Jiawei Qiu [1], Xuandong Sun[1,3], Ziyu Tao [1], Libo Zhang[1,3], Yuxuan Zhou[1], Yuanzhen Chen[3], Weijie Guo [1], Xiayu Linpeng [1], Song Liu [1,4], Wenhui Ren[1], Youpeng Zhong [1,4] ✉, Jingjing Niu [1,4] ✉, Haidong Yuan [2] ✉ & Dapeng Yu [1,4]

Quantum metrology has emerged as a powerful tool for timekeeping, field sensing, and precision measurements in fundamental physics. With the advent of distributed quantum metrology, its capabilities have extended to probing spatially distributed parameters across networked quantum systems. However, scalable implementations of distributed quantum metrology with multiparameter estimation remain limited, particularly due to the challenges of generating and distributing entanglement across a quantum network and dealing with incompatibilities in multiparameter quantum metrology. Here we demonstrate distributed multiparameter quantum metrology on a modular superconducting quantum network with low-loss microwave interconnects, a platform that uniquely combines fast gate operations, adaptive control, and deterministic non-local entanglement generation. Using a control-enhanced sequential protocol, we estimate all three components of a remote vector field, achieving up to 13.72 dB improvement in precision over the individual strategy. We further perform direct estimation of vector field gradients along two directions across spatially separated nodes, realizing a 3.44 dB gain over local entanglement strategies. These results establish superconducting quantum networks as a competitive and reconfigurable platform for scalable multiparameter distributed quantum metrology.

The quest for high-precision measurement is fundamental to scientific advancement. Quantum metrology, which exploits quantum resources such as superposition and entanglement, enables measurement precision beyond classical limits[1–5]. Distributed quantum metrology (DQM) extends this advantage to spatially separated quantum sensors, allowing the characterization of remote or distributed signals[6–15]. It has broad applications from network clock synchronization[16,17] to gravitational and magnetic field mapping[18–21]. Recent demonstrations using photonic[22–29] and atomic[30] platforms have proven that entanglement distributed across nodes can significantly enhance single-parameter sensing. These implementations typically measure global properties, such as an average phase, encoded by mutually commuting generators. Despite the progress, a critical open challenge for practical distributed quantum metrology is the extension to multiparameter

[1]International Quantum Academy, Shenzhen, China. [2]Department of Mechanical and Automation Engineering, The Chinese University of Hong Kong, Shatin, Hong Kong. [3]Southern University of Science and Technology, Shenzhen, China. [4]Shenzhen Branch, Hefei National Laboratory, Shenzhen, China. [5]These authors contributed equally: Jiajian Zhang, Lingna Wang, Yong-Ju Hai, Jiawei Zhang. ✉e-mail: zhongyoupeng@iqasz.cn; niujj@iqasz.cn; hdyuan@mae.cuhk.edu.hk

sensing with non-commuting generators. In this more complex scenario, conventional estimation strategies are highly inefficient[6,13,31–33]. Successfully overcoming this barrier will require two key advancements: first, the generation of high-quality, genuine non-local entanglement across the quantum network, and second, the design of metrological protocols capable of simultaneously estimating multiple parameters with high precision[34–39].

Superconducting circuits offer a powerful platform for DQM, uniquely combining high-speed quantum operations with scalable networking capabilities. Their nanosecond-scale gate times and native compatibility with microwave control and signal transduction[40–43] make them particularly well-suited for detecting fast, weak signals, such as those arising in dark matter and cosmic-ray detection[44–47]. At the same time, advances in superconducting quantum networking have enabled high-fidelity quantum links[48–55] and deterministic entanglement distribution across modular architectures[56], positioning this platform to implement advanced DQM protocols with real-time feedback and dynamic reconfigurability.

In this work, we demonstrate distributed multiparameter quantum metrology in a modular superconducting quantum processor network[56]. Modular architectures provide a scalable path to quantum advantage by interconnecting specialized nodes-dedicated to storage[57–59], processing[60,61], error correction[62–64], or sensing[17,65]-via high-coherence quantum links[56]. Leveraging this framework, we demonstrate multiparameter sensing of a remote 3D vector field using non-local entanglement between a central module and a sensor module, achieving a precision enhancement of up to 13.72 dB in variance over the individual measurement strategy. Furthermore, by creating a distributed 4-qubit GHZ state across two sensor modules via entanglement routing through the central node, we directly estimate the gradients of a spatially varying vector field along two different directions. This yields a 3.44 dB reduction in total variance over strategies with only local entanglement. These results underscore the potential of superconducting modular platforms for building scalable, high-speed, and reconfigurable quantum sensor networks.

## Results
### Experimental setup
The experimental setup, as depicted in Fig. 1, implements a modular superconducting quantum network in a star topology[56]. It comprises multiple sensor modules connected to a central module $\mathcal{A}$ via four 25-cm aluminum coaxial cables, which serve as low-loss transmission lines for microwave photons. Each module hosts four transmon qubits, and the interface between the qubit and cable is equipped with a tunable coupler that enables programmable interaction between qubits and cable modes. The cables function as multimode resonator buses that support standing wave modes, enabling the coherent transfer of microwave photons between modules[56]. By carefully coordinating controls on the qubits and couplers, the system can achieve high-fidelity inter-module operations, with state transfer efficiencies approaching 99%. To generate and distribute entanglement between modules, entangled states are first prepared locally within the central module. The quantum state of one or more qubits is then coherently transferred to remote modules. We then leverage this non-local distributed entanglement to conduct two distinct metrological experiments.

In the first experiment, we employ a maximally entangled state between the central module $\mathcal{A}$ and a single sensor module $\mathcal{B}$ to estimate a remote vector field. The protocol involves first creating a two-qubit entangled state locally on $\mathcal{A}$, then coherently transferring the state of one qubit in this pair to a sensor qubit on module $\mathcal{B}$. This distributed entangled state is subsequently used to perform simultaneous estimation of three independent components of a vector field at the sensor module.

The second experiment uses two sensor modules, $\mathcal{B}$ and $\mathcal{C}$, to estimate multiple spatial gradients of vector fields. The protocol begins by locally preparing a two-qubit entangled state on the central module $\mathcal{A}$, followed by coherently transferring the state of each qubit to two different sensor modules. Local entangling gates are then applied to each sensor module to expand the state into a four-qubit distributed GHZ state. This entangled state is subsequently used to estimate multiple spatial gradients of vector fields applied at the sensor modules. Detailed descriptions of the experimental setup,

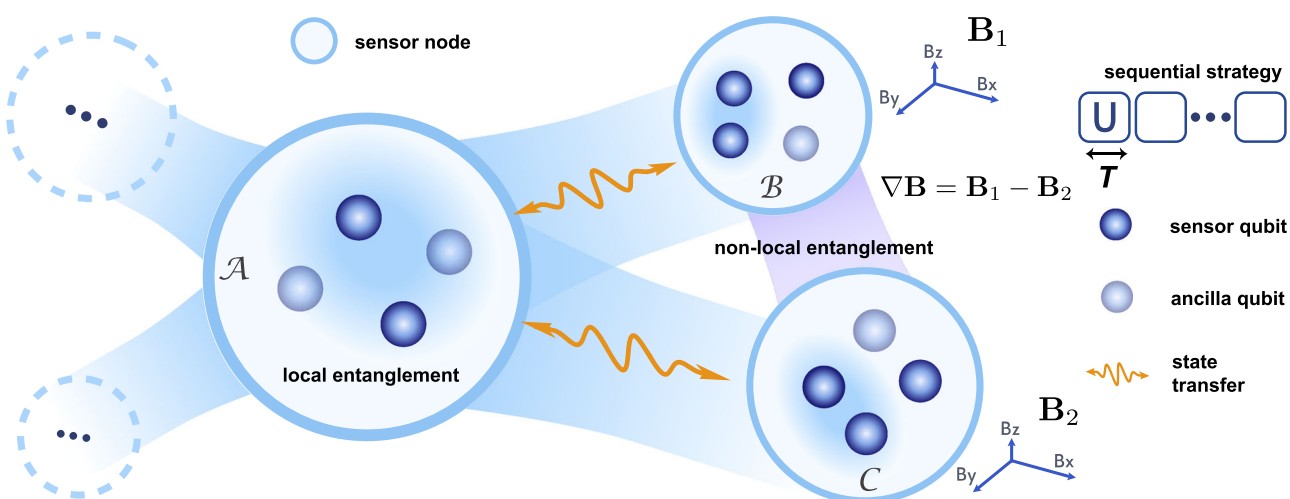

**Fig. 1 | Schematic of distributed quantum metrology with a modular super-conducting sensor network.** The platform comprises a central module $\mathcal{A}$ and multiple spatially separated sensor modules ($\mathcal{B}$, $\mathcal{C}$), each hosting several transmon qubits (blue and light blue spheres). The sensor modules are connected to the central module via low-loss coaxial cables, enabling high-fidelity microwave state transfer (orange arrows) and the generation of non-local entanglement across remote nodes. This architecture supports two key sensing protocols: (1) estimation of remote vector fields encoded by locally non-commuting generators (e.g., $\mathbf{B}_1$, $\mathbf{B}_2$),
and (2) estimation of spatial gradients $\nabla\mathbf{B}$ via distributed entangled probes. Each sensor qubit undergoes a sequence of signal and control unitaries with interrogation time $T$ (top right), implementing a sequential metrological strategy. The blue background indicates local connectivity within a module, while the purple shading highlights modules linked by non-local entanglement. The inclusion of both sensor and ancilla qubits reflects their complementary roles in enabling tailored entangled state preparation, signal encoding, and joint measurement for multi-parameter sensing.

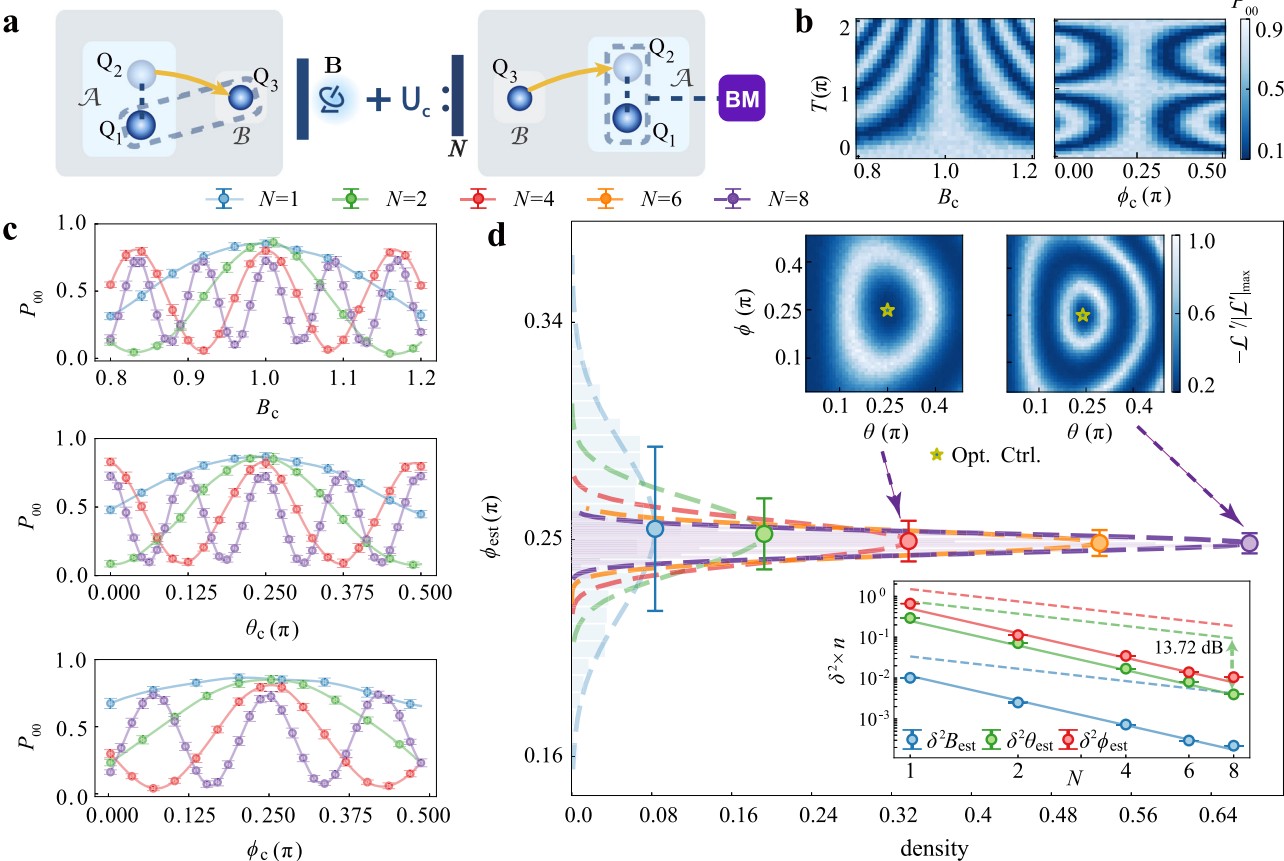

**Fig. 2 | Multi-parameter quantum metrology of a remote magnetic field with the sensor-ancilla network. a** Schematic diagram of the controlled-enhanced sequential strategy. The protocol measures a local magnetic field at the sensor's position using a sequential strategy with one ancilla qubit. The three-component field **B** is parameterized by $B, \theta, \phi$. **b** Measurement probability $P_{00}$ for $N = 8$ cycles as a function of signal encoding time $T$ (from 0 to $2\pi$), scanned across control parameters $B_c$ (left panel) and $\phi_c$ (right panel). **c** Measurement probability $P_{00}$ for a fixed encoding time $T = 1.5\pi$ at different cycle numbers $N = 1, 2, 4, 8$, scanned across control parameters $B_c, \theta_c, \phi_c$. Control parameters not being scanned in (**b**, **c**) are fixed to their optimal values. Error bars denote the standard deviation. **d** Results of parameter estimation. Main panel: Distribution of estimator $\phi_{est}$ for $N = 1, 2, 4, 6, 8$.

Dashed lines represent Gaussian fitting of the estimator histogram; circles and the error bars mark the average value of $\phi_{est}$ and the standard deviation. Upper inset: Landscape of the likelihood function at $N = 4$ (left) and $N = 8$ (right) for parameters $\phi$ and $\theta$. The star marks the optimal control setting that maximizes the likelihood. Lower inset: Assessed precision (variance $\delta^2$) of the three parameters (dots) compared to the $1/N^2$ scaling limits (solid lines) and the $1/N$ scaling bounds of the individual measurement strategy (dashed lines), extracted from $M = 600$ sets of estimators, each derived from $n = 600$ repeats of single-shot measurements. The definition of the error bars is described in Methods. The dashed arrow marks the reduction of $\delta^2\theta_{est}$ relative to its classical bound.

calibration procedures, and implementation protocols are provided in Supplementary Note 2.

For both protocols, we characterize the sensing performance of the distributed network across relevant parameter regimes, and estimate multiple parameters by applying maximum likelihood estimation (MLE) to measurement probabilities obtained from repeated single-shot measurements. This procedure enables a systematic evaluation of the sensing protocol's overall performance, as well as the precision achievable in multiparameter estimation.

**Sensing of remote vector fields**

We first demonstrate the simultaneous estimation of three components of a remote vector field located at one sensor node. In this scenario, we realize a setup where the central module can perform measurements and entangled operations, whereas the sensor module, which interacts with the vector field, has limited capability and can only perform local operations.

The experiment procedure is illustrated in Fig. 2a. We begin by generating a Bell state $|\psi\rangle = \frac{1}{\sqrt{2}}(|00\rangle + |11\rangle)$ in the central module $\mathcal{A}$. One qubit, $Q_1$, is retained on $\mathcal{A}$, while the state of the second qubit, $Q_2$, is transferred to qubit $Q_3$ on the sensor module $\mathcal{B}$, where the vector field is located. This results in a distributed Bell state shared between

$Q_1$ (on $\mathcal{A}$) and $Q_3$ (on $\mathcal{B}$), as detailed in Supplementary Note 2, section 2.3. The sensor qubit $Q_3$ then interacts with the vector field via the signal unitary $U_s(\mathbf{x}) = e^{-i\mathbf{B}\cdot\boldsymbol{\sigma}T}$, where $\mathbf{x} = (B, \theta, \phi)$ are the spherical coordinates of **B** with $\mathbf{B} = (B\sin\theta\cos\phi, B\sin\theta\sin\phi, B\cos\theta)$, $\boldsymbol{\sigma}$ is the vector of Pauli matrices, and $T$ is the signal encoding time. In our experiment, the signal unitary is implemented using a sequence of calibrated quantum gates to accurately emulate field-induced quantum evolution. This approach enables precise and controlled benchmarking of the sensing protocol under well-defined conditions. For practical applications such as microwave sensing, the vector field components may represent physical parameters including field amplitude, phase, and frequency detuning. Each application of the signal unitary is followed by a control unitary $U_c$ applied to the sensor qubit. This sequence is repeated $N$ times, resulting in a total evolution described by $[U_c U_s(\mathbf{x})]^N \otimes I$. To enhance sensing precision, the control operation can be optimized. Theoretically, the optimal control is given by $U_c = U_s^\dagger(\mathbf{x})$[36–39,66]. In practice, as the parameters are initially unknown, the control needs to be implemented adaptively as $U_c = U_s^\dagger(\mathbf{x}_c)$, where $\mathbf{x}_c = (B_c, \theta_c, \phi_c)$ are estimated values of field parameters that are iteratively refined using prior knowledge and accumulated measurement data. After the sensing sequence, the state of the sensor qubit $Q_3$ is transferred back to $Q_2$ in module $\mathcal{A}$ where a projective measurement

on the Bell basis is performed. By repeating the experiment $n$ times, we obtain the probabilities, $\{P_{00}, P_{01}, P_{10}, P_{11}\}$, of the measurement outcomes. Maximum likelihood estimation is then applied to these probabilities to extract estimators of the field parameters. Repeating the entire procedure $M$ times yields $M$ independent estimators, enabling a statistical characterization of the estimation precision. This process can be similarly applied to estimate fields at other sensor modules, and it can be parallelized, facilitating scalable and efficient multiparameter quantum sensing.

In the experiment, the signal parameters are set to $B = 1$, $\theta = \frac{\pi}{4}$ and $\phi = \frac{\pi}{4}$. With optimal control in place, the total evolution simplifies to $U = I \otimes I$, meaning the final state remains identical to the initial Bell state. Consequently, the measured probability peaks at $P_{00}$. In Fig. 2b, we show the profile of $P_{00}$ as a function of the control parameters $B_c$ and $\phi_c$ for different values of the signal encoding time $T$. The width of the central peak around the true parameter values serves as a direct indicator of estimation precision. Notably, for a given circuit sequence depth $N$, the precision for each parameter does not indefinitely improve with increased $T$. The peak width for $B$ narrows, while for $\phi$ it oscillates as $T$ increases. At $T = 1.5\pi$, where the peak width for $\phi$ is the narrowest, we individually scan each control parameter. As shown in Fig. 2c, the central peak narrows consistently for all three parameters as $N$ increases, demonstrating a clear enhancement in estimation precision with greater sequence depth.

To assess measurement precision, we employ maximum likelihood estimation to obtain a set of estimated parameters $\mathbf{x}_{est} = (B_{est}, \theta_{est}, \phi_{est})$. For instance, Fig. 2d displays the distribution of the estimator $\phi_{est}$ for different values of $N$. As the number of cycles $N$ increases, the distribution narrows significantly, demonstrating the enhanced estimation precision achieved through the sequential control strategy. To further characterize the sensor performance throughout the parameter space, we experimentally construct two-dimensional likelihood landscapes by scanning the signal parameters $\theta$ and $\phi$ and recording the corresponding measurement probabilities (see Methods for details). These landscapes visually capture the structure of the likelihood function and illustrate how the estimator sharpens around the true parameter values with increasing $N$, as evidenced by the contraction of the high-likelihood region-consistent with a systematic improvement in estimation precision.

As shown in the lower inset of Fig. 2d, the estimation precision for all three parameters, $\{B_{est}, \theta_{est}, \phi_{est}\}$, exhibits a $1/N^2$ scaling, matching the optimal precision achievable in corresponding single-parameter settings and indicating zero trade-off in simultaneous estimation of the three parameters. The enhancement is enabled by two key features: the use of a GHZ-type entangled probe state, which circumvents limitations imposed by locally non-commuting generators, and the adaptive control-enhanced sequential strategy, which amplifies precision through repeated signal-control interrogation cycles. For comparison, we benchmark these results against an individual measurement strategy that employs separable probes and measurements, where the total resources are divided evenly to estimate each parameter individually. The achieved maximum precision gains, in terms of variance, are 12.8 dB for $B_{est}$, 13.72 dB for $\theta_{est}$, and 12.56 dB for $\phi_{est}$.

## Distributed sensing of vector field gradients

In the second experiment, we realize a scenario where each sensor module is capable of performing local entangling operations and measurements. Our objective is to implement a distributed gradiometric protocol. Here, two sensor modules are exposed to distinct vector fields, $\mathbf{B}_1$ and $\mathbf{B}_2$, and we simultaneously estimate all components of the gradient $\nabla\mathbf{B} = \mathbf{B}_1 - \mathbf{B}_2$.

We begin by preparing a non-local entangled (NLE) probe state $|\Psi_0\rangle = (|0011\rangle - |1100\rangle)/\sqrt{2}$ across sensor modules $\mathcal{B}$ and $\mathcal{C}$. The protocol starts with generating a Bell pair between qubits $Q_1$ and $Q_2$ in the

central module $\mathcal{A}$, followed by coherent state transfer from $Q_1$ to $Q_3$ on $\mathcal{B}$ and from $Q_2$ to $Q_5$ on $\mathcal{C}$. Local CNOT gates are then applied within each sensor modules to extend the entanglement, resulting in a four-qubit GHZ state. The final probe state $|\Psi_0\rangle$ is obtained by applying additional X gates to $Q_3$ and $Q_4$, and a Z gate to $Q_5$. The overall preparation fidelity is measured to be 76.16%.

The entangled probe state then interacts with local vector fields $\mathbf{B}_1$ and $\mathbf{B}_2$. The evolution over time $T$ is governed by the operator $U_S(\mathbf{B}_1, \mathbf{B}_2) = U_{s1}^{\otimes 2} \otimes U_{s2}^{\otimes 2}$, where $U_{sj} = e^{-i\mathbf{B}_j \cdot \sigma T}$ for $j = 1, 2$. This evolution can be reparametrized as $U_S(\nabla\mathbf{B}, \Sigma\mathbf{B})$, where $\nabla\mathbf{B} = \mathbf{B}_1 - \mathbf{B}_2$ represents the gradient, and $\Sigma\mathbf{B} = \mathbf{B}_1 + \mathbf{B}_2$ represents the sum. After each signal evolution, a control operation $U_C$ is applied. The cycle is repeated $N$ times, resulting in the total evolution $[U_C U_S(\nabla\mathbf{B}, \sum\mathbf{B})]^N$. The control operation is implemented adaptively as $U_C = U_S^\dagger(\nabla\mathbf{B}_C, \sum\mathbf{B}_C)$ where $(\nabla\mathbf{B}_C, \sum\mathbf{B}_C)$ are the iteratively updated estimates of $(\nabla\mathbf{B}, \sum\mathbf{B})$ based on accumulated measurement outcomes. Finally, Bell measurements are performed on both modules $\mathcal{B}$ and $\mathcal{C}$. The experiment is repeated $n$ times to obtain the probability distribution $\{P_{ijkl}\}$, where $i, j, k, l \in \{0, 1\}$. From these probabilities, the gradients $\nabla\mathbf{B}_{est} = (\nabla B_x, \nabla B_y, \nabla B_z)$ are inferred via maximum likelihood estimation.

To validate this protocol, we examine two representative configurations for the vector fields $\mathbf{B}_1$ and $\mathbf{B}_2$. The first is $\mathbf{B}_1 = \mathbf{B}_2 = \sqrt{2}/4(1, 1, 0)$, corresponding to two vector fields confined to the XY-plane with a known zero Z-component. The second is a fully three-dimensional case with $\mathbf{B}_1 = \mathbf{B}_2 = 1/2(1, 1, \sqrt{2})$. We assume $\mathbf{B}_1 = \mathbf{B}_2$ in both cases without loss of generality, since any initial mismatch $\mathbf{B}_1 \neq \mathbf{B}_2$ can be iteratively corrected by applying an adaptive compensation $\nabla\mathbf{B}_{est}$ to one of the nodes. This procedure asymptotically aligns the two fields and does not affect the ultimate estimation performance[67] (see Supplementary Note 1, Section 1.3.1). For both configurations, experiments are conducted for $N = 1$ to 4 cycles. As shown in Fig. 3c, the observed precision approaches the theoretical limit and exhibits the expected $1/N^2$ scaling. Further increases in $N$ are currently constrained by decoherence and control errors. We quantitatively analyze the impact of these noise sources and their contribution to deviations from the theoretical limit in Supplementary Note 3, Section 3.3, and note that such limitations can be mitigated through future hardware improvements.

Under optimal control unitary, the probabilities of the measurement results are dominated by $P_{0010}$ and $P_{1000}$ (see Supplementary Note 2, Section 2.3). Taking the $y$-component estimator $\nabla B_{y_{est}}$ for the 2D gradient case as an example, its normalized distribution in Fig. 3b shows that as $N$ increases from 1 to 4 at $T = 1.5\pi$, the standard deviation progressively decreases, indicating improved estimation precision. Similarly, the precision for the $x$-component, $\delta\nabla B_{x_{est}}$, follows an identical trend, see Supplementary Note 2 (Section 2.3) and Supplementary Note 3 (Section 3.2).

For comparison, we also perform an experiment using only local entanglement within each module (see Fig. 4a, b). In this reference scheme, both qubits in each sensor module serve as local sensors. Bell states are prepared independently within modules $\mathcal{B}$ and $\mathcal{C}$ to estimate $\mathbf{B}_1$ and $\mathbf{B}_2$ separately. In each module, the signal encoding is interleaved with optimal control operations, and Bell measurements are performed after the evolution. The gradient $\nabla\mathbf{B}$ is then computed by differencing the two local estimates. We use $g_{LE/NLE} = 10\log_{10}[(\sum_{i \in \{x,y\}} \delta^2 B_{i_{est}})_{LE}^{exp(sim)}/(\sum_{i \in \{x,y\}} \delta^2 B_{i_{est}})_{NLE}^{exp}]$ to quantify the advantage provided by distributed sensing with non-local entanglement (NLE) over the local sensing with local entanglement (LE). Similar comparisons between local and non-local strategies have also been made in prior works[23,68]. Here, we restrict the comparison to the two-component gradient, because in the three-component case, the quantum Fisher information matrix for the local strategy becomes singular, rendering the estimation infeasible for the local sensing (see Supplementary Note 1, Section 1.3.2). We assess the gain in the

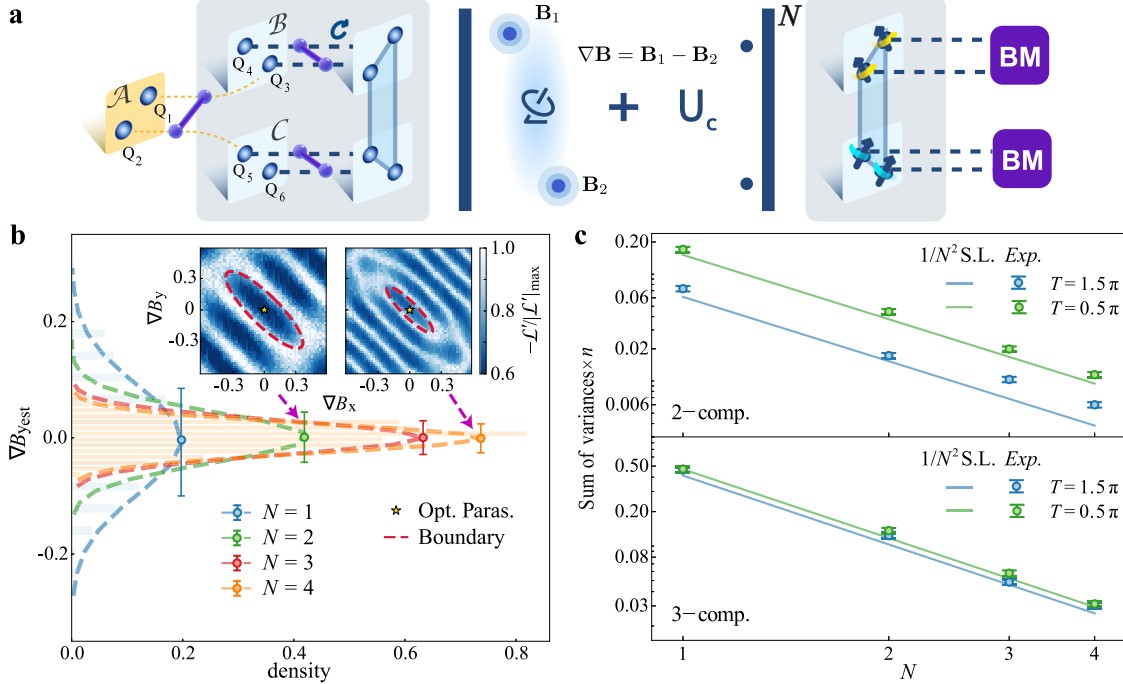

**Fig. 3 | Multi-parameter quantum metrology of the magnetic field gradient.**
**a** Distributed sensing scheme for directly estimating the gradients of two vector fields with a non-local entangled state. The left section shows the probe state initialization. The system consists of a central module (yellow box) and two sensor modules (light blue boxes within the gray area). Blue spheres within the modules represent qubits, the purple stick-ball model represents CNOT gates, and the circular arrow denotes a local rotation. Four sensor qubits are initialized in a non-local entangled state, $|\Psi_0\rangle = \frac{1}{\sqrt{2}}(|0011\rangle - |1100\rangle)$. The middle section illustrates the signal-control encoding process, while the right section depicts simultaneous Bell measurements on the two encoded sensor modules. Sensor qubits in the same colored stripes are encoded with identical vector field signals. **b** Benchmarking the performance of the gradiometer. Main panel: Distribution of estimator $\nabla B_{y_{est}}$ for

$N = 1$ to $N = 4$. Dashed lines represent the Gaussian fit of the estimator histograms, while circles and error bars indicate the mean values and standard deviations ($\delta \nabla B_{y_{est}}$). Inset: Landscape of the log-likelihood function $\mathcal{L}'$ at $N = 2$ (left) and $N = 4$ (right) (see Methods). The star marks the optimal control point and the red dashed contour denotes the parameter-estimation region. **c.** The gradient estimation precision, evaluated by the sum of variances, $\sum_{i \in \{x,y,z\} \cup \{x,y\}} \delta^2 \nabla B_{i_{est}}$, obtained from $M$ sets of estimators ($M = 600$), each derived from $n$ measurement shots ($n = 600$). Top panel: two-component vector field. Bottom panel: three-component vector field. Solid lines indicate the $1/N^2$ scaling limit ($1/N^2$ S.L.). Precision estimated at $T = 0.5\pi$ is marked in green, while precision at $T = 1.5\pi$ is marked in blue. The definition of the error bars is described in Methods.

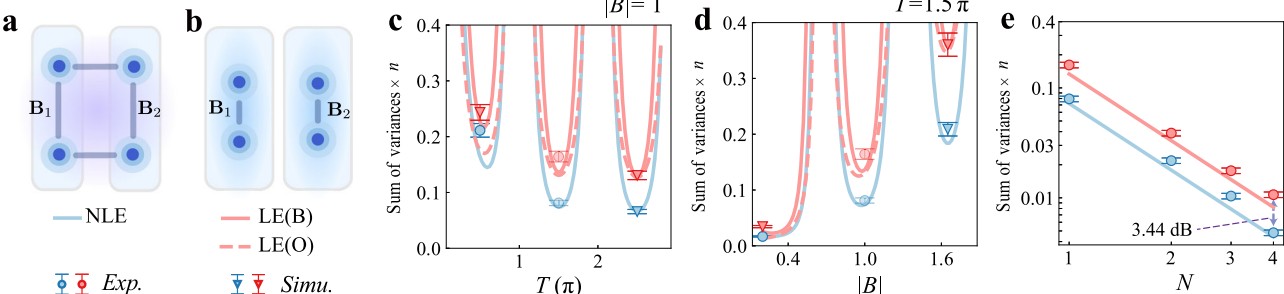

**Fig. 4 | Strategies comparison for gradient estimation of a 2-component vector field. a**, **b** Schematic diagrams of different strategies: **a** Distributed sensing with non-local entanglement (NLE); **b** Sensing with local entanglement (LE). **c**–**e** Comparison of the precision ($\sum_{i \in \{x,y\}} \delta^2 \nabla B_{i_{est}}$) of the two strategies. **c** Precision as a function of field strength $|\mathbf{B}|$ for the three strategies at $T = 1.5\pi$ and $N = 1$. **d** Precision as a function of encoding time $T$ at $|\mathbf{B}| = 1$ and $N = 1$. **e** Precision versus number of cycles $N$ for the two strategies at $T = 1.5\pi$ and $|\mathbf{B}| = 1$. The solid curve

represents the theoretical precision bound for the local entanglement strategy with the probe state and measurement taken as the Bell state and Bell measurement, labeled as LE(B). The dashed curve represents the theoretical precision bound for the local entanglement strategy with the optimal probe state and optimal measurement, labelled as LE(O). The definition of the error bars in (**c**–**e**) is described in Methods.

two-component case under several conditions: first, by fixing the field amplitude at $|\mathbf{B}| = 1$ and sweeping the encoding time $T$ from $0.5\pi$ to $2.5\pi$ (Fig. 4c); next, by fixing $T = 1.5\pi$ and scanning $|\mathbf{B}|$ over the range 0.2 to 1.65 (Fig. 4d); and finally, by holding $|\mathbf{B}| = 1$ and $T = 1.5\pi$ constant while varying the number of sequential encoding steps $N$. For $N = 4$, the distributed protocol achieves a maximal gain of 3.44 dB over the local strategy.

## Discussion
By deterministically generating high-fidelity distributed entanglement across network nodes, our experiment demonstrates an inverse-square scaling of the variance with sensing circuit depth for the estimation of all three components of a remote vector field, with an improvement of up to 13.72 dB over an individual strategy. In the context of gradient estimation, our distributed strategy–enabled by

modular superconducting architecture, distributed entanglement, and control-enhanced sequential strategies—surpasses local strategies, achieving a 3.44 dB reduction in total variance when estimating gradients along two distinct directions in distributed two-dimensional vector fields. The four-qubit GHZ state between two non-nearest-neighbor nodes used in the gradient sensing exhibits a fidelity of 80.36%, which is among the highest demonstrated across physically separated network nodes to date[56,69–71] and sufficient to realize a clear advantage of the distributed protocol.

These results establish a scalable framework for distributed quantum sensing of vector fields and their gradients with enhanced precision and architectural flexibility. Our approach offers a concrete path toward quantum-enhanced sensor networks that can be applied to a wide range of practical scenarios, including electromagnetic field monitoring, navigation, and remote detection. Looking forward, the integration of adaptive control, error correction, and expanded network topologies could unlock new frontiers in precision sensing and real-time quantum signal processing.

## Methods

### Experimental platform

We implement the distributed quantum sensor on a modular superconducting quantum network consisting of five modules[56]. Each module hosts four capacitively coupled transmon qubits, enabling local entangled operations. Inter-module communication is realized via high-fidelity quantum state transfer through low-loss microwave links (four 25-cm aluminum coaxial cables). Tunable couplers at each cable-qubit interface allow programmable interaction between the qubits and the multimode cable resonators, supporting coherent photon transfer across modules. The entangled probe states are prepared through a combination of quantum state transfer, local CNOT gates, and single-qubit rotations (see Supplementary Note 2, Section 2.3). The vector field signal and control unitaries are digitally simulated using a $U(3)$ formalism (see Supplementary Note 2, Section 2.2), where the signal parameters are set to span representative conditions, while control parameters are set according to prior experimental calibration. To extract information about the encoded signals' parameters, Bell measurements are performed on selected qubits to obtain the output state probabilities. Maximum likelihood estimation is then used to reconstruct the parameters of interest from the measurement data, enabling quantitative evaluation of sensor precision and performance.

### Maximum likelihood estimation for remote field sensing

To estimate the unknown signal parameters, we employ maximum likelihood estimation (MLE) based on experimentally acquired measurement outcomes[72]. For a fixed control configuration $\mathbf{x}_c$, we implement the sensing circuit and perform $n$ repeated measurements on a fixed basis (e.g., the Bell basis), yielding outcomes $y_i \in \{00, 01, 10, 11\}$ ($j = 1, \cdots, n$). Let $n_i$ be the count of outcome $i \in \{00, 01, 10, 11\}$ and $P_i^{\exp} = n_i/n$ the corresponding empirical frequency. Denote by $P_i^{\text{ideal}}(\mathbf{x}, \mathbf{x}_c)$ the ideal model-predicted probability of outcome $i$ under parameter $\mathbf{x}$ and control $\mathbf{x}_c$. For clarity, we first define the likelihood of the full measurement record

$$\mathcal{L}(y_1, \ldots, y_n \mid \mathbf{x}, \mathbf{x}_c) = \prod_{j=1}^{n} P^{\text{ideal}}(y_j \mid \mathbf{x}, \mathbf{x}_c). \quad (1)$$

Grouping identical outcomes, we define the normalized log-likelihood objective as

$$\begin{aligned} \mathcal{L}(\mathbf{x}, \mathbf{x}_c) &\equiv \frac{1}{n} \ln \mathcal{L}(y_1, \ldots, y_n \mid \mathbf{x}, \mathbf{x}_c) \\ &= \sum_i P_i^{\exp} \ln \left( P_i^{\text{ideal}}(\mathbf{x}, \mathbf{x}_c) \right). \end{aligned} \quad (2)$$

The estimated parameters are obtained by maximizing the normalized log-likelihood function: $\mathbf{x}_{\text{est}} = \arg\max_{\mathbf{x}} \mathcal{L}(\mathbf{x}, \mathbf{x}_c)$, using gradient-based

optimization. To quantify the estimation precision, we perform $M$ independent realizations of the full estimation procedure to obtain $M$ independent estimators for each parameter, from which the variance $\delta^2$ is evaluated; the error bars of the plotted variances in Figs. 2–4 are then given by its standard deviation $\text{SD}(\delta^2) = \sqrt{2/(M-1)}\delta^2$[73].

Since the optimal control parameters depend on the true signal parameters, we implement an adaptive protocol to iteratively refine $\mathbf{x}_c$[67,74–76]. This process consists of four steps. (1) Initialization: A set of control parameters, $\mathbf{x}_c^{(0)} = (B_c^{(0)}, \theta_c^{(0)}, \phi_c^{(0)})$, is chosen (e.g., randomly or based on prior knowledge). The sensing circuit is executed, yielding the empirical probability distribution $\{P_i^{\exp(0)}\}$. (2) First estimation: The first parameter estimate is obtained via MLE: $\mathbf{x}_{\text{est}}^{(1)} = \arg\max_{\mathbf{x}} \sum_i P_i^{\exp(0)} \ln(P_i^{\text{ideal}}(\mathbf{x}, \mathbf{x}_c^{(0)}))$. This estimate is then used to update the control parameters $\mathbf{x}_c^{(1)} = \mathbf{x}_{\text{est}}^{(1)}$. (3) Iteration: Using the updated control parameters $\mathbf{x}_c^{(1)}$, the experiment is repeated to obtain a new empirical distribution $P_i^{\exp(1)}$. (4) Joint estimation: The estimator and control parameters are updated by maximizing the joint log-likelihood incorporating all data collected so far: $\mathbf{x}_c^{(2)} = \mathbf{x}_{\text{est}}^{(2)} = \arg\max_{\mathbf{x}} \sum_i P_i^{\exp(0)} \ln(P_i^{\text{ideal}}(\mathbf{x}, \mathbf{x}_c^{(0)})) \times \sum_i P_i^{\exp(1)} \ln(P_i^{\text{ideal}}(\mathbf{x}, \mathbf{x}_c^{(1)}))$. This adaptive process is repeated for $K$ cycles. The final estimate after $K$ iterations is: $\mathbf{x}_{\text{est}}^{(K)} = \arg\max_{\mathbf{x}} \mathcal{L}_{\text{joint}}$, where $\mathcal{L}_{\text{joint}} = \prod_{m=1}^{K-1} \sum_i P_i^{\exp(m)} \ln [P_i^{\text{ideal}}(\mathbf{x}, \mathbf{x}_c^{(m)})]$ is the joint likelihood function[77]. The $K$-th MLE considers all $K-1$ experiment results, the control parameters will be updated with the increasing of iteration cycles. After at most $R_{\text{iter}} = 40$ rounds, the estimators $B_{\text{est}}$, $\theta_{\text{est}}$, and $\phi_{\text{est}}$ converge to stable values.

We demonstrate the convergence of the adaptive protocol in Fig. 5 by minimizing the joint likelihood function $-\mathcal{L}_{\text{joint}}$ in each cycle and tracking the evolution of the cost function $(-1)^{R_{\text{iter}}} \times \mathcal{L}_{\text{joint}}$. The convergence process is shown to be robust both across various initial guesses at $N = 1$ and across sequential copies ($N = 1$ to $6$) for a fixed initial guess, as depicted in Fig. 5a, b. Numerical analysis of the joint likelihood function at $N = 4$ (Fig. 5c) indicates that, while the landscape initially appears irregular, it evolves into an optimal configuration similar to that in Fig. 2b after a few cycles, highlighting the effectiveness of this adaptive optimization process.

### Experimental likelihood benchmark

To assess the sensitivity and precision of our sensing protocol under a given control setting, we perform a likelihood function benchmark. It is crucial to distinguish this from the parameter estimation procedure: this benchmark is not used to extract unknown parameters. Instead, its purpose is to visualize the structure of the likelihood landscape and validate the agreement between our experimental results and the theoretical model.

This benchmarking method operates as follows: We fix the control parameters $\mathbf{x}_c$ to their optimal values and then scan the signal parameters $\mathbf{x}$ over a local grid. For each point on this grid, we run the sensing experiment and record the resulting measurement probability distribution $P_i^{\exp}(\mathbf{x}, \mathbf{x}_c)$. We then construct the experimental log-likelihood function $\mathcal{L}' = \sum_i P_i^{\exp} \ln(P_i^{\exp}(\mathbf{x}, \mathbf{x}_c))$ by comparing these probabilities to the reference distribution $P_i^{\exp}$ measured at the true signal parameter values.

The 2D landscapes of the normalized $\mathcal{L}'$, shown in Figs. 2d and 3b, reveal the parameter estimation space. This space is characterized by an internal boundary whose saddle point corresponds to the alignment of the guess signal parameters with the true values. The contraction of this boundary with an increasing number of cycles $N$ visually demonstrates the enhanced precision achieved through our sequential strategy.

In summary, this benchmark provides an intuitive, fully data-driven method to evaluate sensing performance; it is strictly a diagnostic tool and does not influence the adaptive protocol or the final

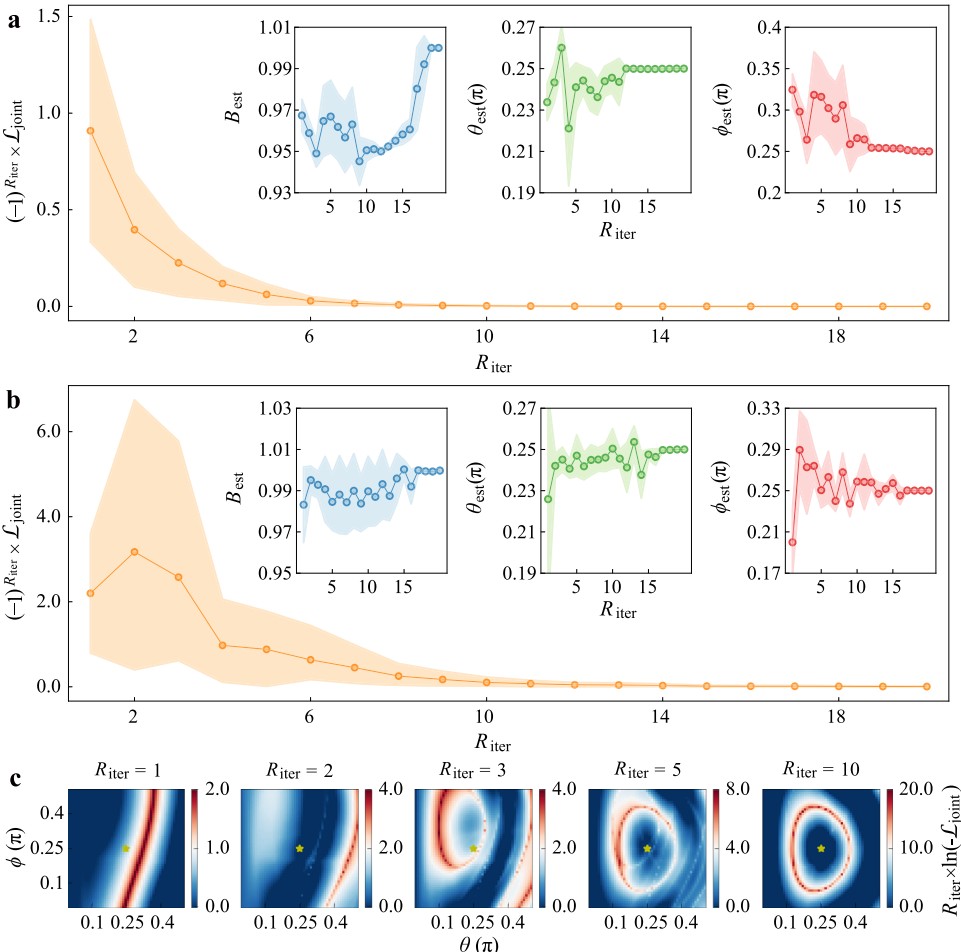

**Fig. 5 | Adaptive control-enhanced metrology for simultaneous three-parameter estimation with signal parameters** $(B, \theta, \phi) = (1, \frac{\pi}{4}, \frac{\pi}{4})$**.** Here, $R_{iter}$ denotes the iteration index of the adaptive estimation-control loop. **a** The results of adaptive iterations starting from different initial guess values (10 sets randomly chosen within the boundary of the landscape) with $N = 1$. **b** The results of adaptive

iterations with different sequential copies $N = 1$ to 6 and fixed initial guess (1 set randomly chosen within the boundary of the landscape for $N = 1$). **c** The calculated likelihood function landscape for $N = 4$ after iteration cycles 1, 2, 3, 5, 10. Stars indicate the locations of the optimal control parameters.

estimation outcome. The close agreement between our experimentally probed landscapes and numerical simulations validates the underlying model used in our actual MLE procedure (see Supplementary Note 3, Section 3.1). This paves the way for developing a programmable sensor system reliant solely on direct experimental measurements and parameter feedback, with the benchmark serving as a crucial verification step.

## Ultimate precision for remote vector field sensing

The Hamiltonian for the sensor qubit can be expressed as $H = \mathbf{B} \cdot \boldsymbol{\sigma}$, where the vector field $\mathbf{B} = (B_x, B_y, B_z)$ can be expressed in the spherical coordinates as $B_x = B \sin\theta \cos\phi$, $B_y = B \sin\theta \sin\phi$, and $B_z = B \cos\theta$ with $\mathbf{x} = (B, \theta, \phi)$ as the parameters in the spherical coordinates. The unitary operator generated under the free evolution over time $T$ is given by $U_s(\mathbf{x}) = e^{-i\mathbf{B} \cdot \boldsymbol{\sigma} T}$.

To evaluate the performance of a sensing strategy, we employ the quantum Cramér-Rao bound, given by $n \operatorname{Cov}(\mathbf{x}_{est}) \geq F_Q^{-1}$, where $\operatorname{Cov}(\mathbf{x}_{est})$ is the covariance matrix of the estimators for the unknown parameters $\mathbf{x}$, $n$ is the number of measurement repetitions, and $F_Q$ is the quantum Fisher information matrix (QFIM). The overall estimation precision for multiple parameters is quantified by the total variance ($\operatorname{Tr}[\operatorname{Cov}(\mathbf{x}_{est})]$).

For the optimally controlled strategy, the optimal control $U_c = U_s^\dagger$ is applied after each signal unitary $U_s$, and the sequence is repeated $N$

times[36–39,66]. In this case, the quantum Fisher information matrix is given by $F_Q^{max} = 4N^2 \begin{pmatrix} T^2 & 0 & 0 \\ 0 & \sin^2(BT) & 0 \\ 0 & 0 & \sin^2(BT)\sin^2\theta \end{pmatrix}$. In Supplementary Note 1 (Section 1.2), we also explicitly compute the classical Fisher information matrix (CFIM) under the projective measurements in the Bell basis, which matches the QFIM, confirming that the quantum Cramér-Rao bound can be saturated. The ultimate precision limits for the three parameters are then given by $n(\delta B_{est})^2 \geq \frac{1}{4N^2 T^2}$, $n(\delta\theta_{est})^2 \geq \frac{1}{4N^2 \sin^2(BT)}$, $n(\delta\phi_{est})^2 \geq \frac{1}{4N^2 \sin^2(BT)\sin^2\theta}$, where $(\delta x_{est})^2$ denotes the variance of the estimator and $n$ is the number of measurement repetitions. For each of the parameters, this is also the highest precision that can be achieved in the single-parameter case where the other two parameters are taken as known values. The optimal strategy thus achieves the highest precision for all three parameters simultaneously without any tradeoff.

This is compared with the individual measurement strategy[39], where the total number of channel uses $N$ is divided evenly among three groups, and each group is dedicated to estimating one parameter independently. This strategy exhibits a $1/N$ scaling with respect to the number of channel uses as $n(\delta B_{est})^2 \geq \frac{3}{4NT^2}$, $n(\delta\theta_{est})^2 \geq \frac{3}{4N\sin^2(BT)}$, $n(\delta\phi_{est})^2 \geq \frac{3}{4N\sin^2(BT)\sin^2\theta}$. This serves as a baseline for assessing the enhancements provided by the distributed strategy.

**Precision limit for gradient estimation with non-local entanglement**

Under the optimal controlled sequential scheme, the total dynamics is given by $(U_C U_S)^N$, where $U_S(\mathbf{x}) = U_{s1}^{\otimes 2} \otimes U_{s2}^{\otimes 2}$, $U_C = U_S^\dagger(\mathbf{x}_c)$ with $\mathbf{x}_c = (\nabla B_c, \sum B_c)$ as the adaptively updated estimate of $\mathbf{x} = (\nabla B, \sum B)$[36,39]. As detailed in the Supplementary Note 1 (Section 1.3.1), using a non-local entangled probe state $|\Psi_0\rangle = (|0011\rangle - |1100\rangle)/\sqrt{2}$, the QFIM for the simultaneous estimation of $\mathbf{x}$ is given by $F_Q = N^2 \begin{pmatrix} F_- & \mathbf{0} \\ \mathbf{0} & F_+ \end{pmatrix}$, where $F_-$ and $F_+$ correspond to the QFIMs for estimating $\nabla B$ and $\sum B$, respectively. We further verify that under local projective measurements in the Bell basis on each sensor module, the classical Fisher information matrix coincides with the QFIM, confirming that the quantum Cramér-Rao bound can be saturated by the measurement strategy. The block diagonal form of the QFIM implies that the precision of estimating the gradients $\nabla B$ is not affected by $\sum B$. For the estimation of $\nabla B_x$ and $\nabla B_y$ of two-dimensional vector fields, we have

$$n\left[\left(\delta \nabla B_{x_{est}}\right)^2 + \left(\delta \nabla B_{y_{est}}\right)^2\right] \geq \frac{1}{4N^2}\left(\frac{1}{T^2} + \frac{B^2}{\left(1+3\sin^2(BT)\right)\sin^2(BT)}\right), \quad (3)$$

where $B$ is the magnitude of the vector field with $B = \sqrt{B_x^2 + B_y^2 + B_z^2}$. For the analysis of the three-dimensional case, see Supplementary Note 1, Section 1.3.1.

**Precision limit for gradient estimation with local entanglement**

We theoretically determine the precision limits for the estimation of the gradients under the local strategy. In this strategy, local entangled states are employed to first estimate the vector field at each sensor module separately; the gradients are then obtained by computing the differences. Under this strategy, the precision for estimating $\nabla B_x$ and $\nabla B_y$ of two-dimensional vector fields is given by

$$n\left[\left(\delta \nabla B_{x_{est}}\right)^2 + \left(\delta \nabla B_{y_{est}}\right)^2\right] \geq \frac{1}{8N^2}\left(\frac{1}{T^2} + \frac{B^2}{\sin^2(BT)}\right). \quad (4)$$

This, however, requires an optimal probe state that depends on the true values of the parameters, which is unknown a priori. An adaptive preparation is thus needed. For practical implementation, we use the Bell state $\frac{1}{\sqrt{2}}(|00\rangle + |11\rangle)$ instead, which is parameter-independent and does not require adaptive preparation. In this case, the precision is given by

$$n\left[\left(\delta \nabla B_{x_{est}}\right)^2 + \left(\delta \nabla B_{y_{est}}\right)^2\right] \geq \frac{1}{8N^2\sin^2(BT)}\left(\frac{B^2 - \cos^2(BT)B_x^2}{T^2B_x^2} + \frac{B^2}{\sin^2(BT)}\right). \quad (5)$$

When $B_x = B_y$, as the case in the experiment, the two bounds become almost the same near the optimal time point with $\sin(BT) = 1$.

In both cases, the corresponding precision bound can be saturated by performing projective measurements in the Bell basis on each sensor module. We explicitly calculate the CFIM under the projective measurement of the Bell-basis, confirming the optimality of the measurement strategy in our system. The full derivation is provided in Supplementary Note 1, Section 1.3.2.

## Data availability

The data that support the plots within this paper and other findings of this study are available from the corresponding author upon request.

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

## Acknowledgements

We thank Xiu-Hao Deng, Zhibo Hou, and Raphael Kaubruegger for insightful discussions. This work was supported by the National Natural

Science Foundation of China (12374474 and 12174178), the Quantum Science and Technology-National Science and Technology Major Project (2021ZD0301703), the Science, Technology and Innovation Commission of Shenzhen Municipality (KQTD20210811090049034, RCBS20231211090824040, and RCBS20231211090815032), the Shenzhen-Hong Kong Cooperation Zone for Technology and Innovation (HZQB-KCZYB-2020050), Guangdong Basic and Applied Basic Research Foundation (2024A1515011714 and 2022A1515110615), Research Grants Council of Hong Kong (14309223, 14309624, and 14309022), the Guangdong Provincial Quantum Science Strategic Initiative (GDZX2303007), and the Department of Science and Technology of Guangdong Province (2020B0303050001).

## Author contributions

J.N. initiated the project and designed the experiment. J.J.Z. conducted the measurements and analyzed the data with Y.-J.H. under the supervision of J.N. L.W. and Y.-J.H. provided theoretical support guided by H.Y. J.W.Z. developed the microwave electronics infrastructure with assistance from X.S. L.Z. fabricated the devices with support from Y.X.Z. and S.L. W.H., J.Q., and Y.L. contributed to the experimental setup. J.C., J.J., Z.T., Y.C., X.L., W.G., and W.R. participated in discussions of the results. J.J.Z., L.W., Y.-J.H., J.N., and H.Y. wrote the manuscript with input from all authors. H.Y., J.N., Y.P.Z., and D.Y. supervised the project.

## Competing interests

The authors declare no competing interests.
