## [Transparent Peer Review file · Nature Communications]

Distributed multi-parameter quantum metrology with a superconducting quantum network

Corresponding Author: Dr Haidong Yuan

Version 0:

Reviewer comments:

Reviewer #1

(Remarks to the Author)

The authors experimentally perform two different multi-parameter quantum sensing tasks to greater than standard quantum limit precision using a network of superconducting qubit sensors:

- (1) estimate all three components of a "remote" vector field located at one sensor node (i.e. they use an ancilla). In this setting it is not strictly necessary that the network be "distributed" but the motivation is a setting where one can do measurements and entangled operations in some central module and only local operations and sensing with the sensor qubit.
- (2) estimate all components of a gradient of a magnetic field between two qubit sensors

The paper provides a significant experimental demonstration of a toy, yet genuinely multiparameter, quantum sensing task using a small network of sensors and, in my opinion, could reasonably merit publication in Nature Communications provided certain things are addressed. Some general comments:

- It is unclear what exactly is being compared in the "sensing of remote vector fields" section when the authors claim a ~6-7dB gain for estimating the parameters. Presumably, the standard quantum limit in terms of the number of cycles N . In the abstract this is referred to as the "classical strategy" but the strategy is not classical as the sensors are fundamentally qubits. It would be more correct to refer to a "local" or "unentangled" strategy as the sensors are quantum mechanical, whether or not entanglement is used. However, it is actually unclear to me if "local" or "unentangled" is the correct characterization or if it should be comparing to using the same (entangled) states, but measuring after each cycle N ? One thing that might help is to include the standard quantum limit that is being compared to on the plots, similar to how the $LE(B)$ and $LE(O)$ results are compared in Fig. 4 e
- In "the distributed sensing of vector field gradients" the authors claim that without loss of generality they can take the magnetic fields B_1 and B_2 to be equal (implying that the gradient being estimated is zero) as "we can adaptively add $\nabla \vec{B}_{est}$ at site 2, which makes them asymptotically equal and does not affect the asymptotic performance. This is not obvious as the associated estimate would have to be to Heisenberg precision to make them asymptotically equal. While I expect that such an adaptive scheme is achievable it seems somewhat non-trivial to me. A reference or two and some further discussion seems necessary here.
- The summary section is a bit sparse and could use a little more detailed and specific discussion of implications and outlook. Furthermore, the authors claim "through the deterministic generation of high-fidelity entanglement..." Is ~70-80% fidelity truly "high fidelity" in this context? I am a theorist so it may very well be a justified claim, but this might be a place where some more specific comparison with other systems and outlook might be helpful.

Additional minor comments:

- In the paragraph right above "Experimental setup" on pg. 2 the authors refer to a "central modular" and a "sensor modular". I believe this should read "central module" and "sensor module". I.e. the system as a whole is modular, but the individual parts are modules.

Reviewer #2

(Remarks to the Author)

The manuscript presents an experimental implementation of the sequential quantum metrology protocol introduced in [PRL 117, 160801 (2016), Ref. 58]. In short, the protocol relies on the idea that by interspersing a sequence of gates, each encoding a set of parameters, with control operations that effectively cancel the encodings, the ultimate Heisenberg limits can be reached even when dealing with multiple parameters (multiparameter metrology) generated by non-commuting Hamiltonians.

The protocol has been previously implemented on a photonic platform, first without the need to use entanglement in the single-parameter setting [PRL 123, 040501 (2019), Ref. 59], and later in the three-parameter (3D vector estimation) setting [Sci. Adv. 7, eabd2986 (2021), Ref. 57], in which entanglement between sensing and ancillary qubits (polarisation and path d.o.f.s of a single photon) was necessary to attain the multiparameter Heisenberg limit.

The current work is based on an experiment involving a different platform consisting of 5 interconnected superconducting chips (modules), each comprising 4 qubits, designed for quantum computing applications. It extends the aforementioned previous implementations of the protocol. In particular, it firstly repeats the scheme of [Sci. Adv. 7, eabd2986 (2021), Ref. 57] of sensing a 3D vector of parameters, but thanks to the usage of a modular quantum computing platform, the sensing qubit, after being entangled with an ancillary one within one chip, is transferred to another module where the encoding (and control) is performed before being transferred back again, so that a Bell measurement within the original module can be performed. The Authors then propose and implement a new version of the protocol, which they show to be suboptimal but still surpassing all classical strategies. Its task is to estimate the difference between the 3D vectors encoded at two superconducting modules, which is achieved by preparing a 4-qubit GHZ-like state involving two qubits in each of the two modules and performing Bell measurements at each module. Because both schemes involve entanglement between different modules, the Authors present them as examples of distributed quantum metrology.

Although I find the achieved results very impressive, I cannot recommend the work to be published in Nature Communications for the following reasons:

- a) The concept and motivation behind the work has already been achieved in the aforementioned works on the photonic platform.
- b) The superconducting platform used is designed for quantum computing and, as I cannot really view it as sensor (network), I find the presented results more as a simulation of the quantum metrology protocol of [PRL 117, 160801 (2016), Ref. 58], which I doubt to be applicable to any real-life sensing task with external parameters to be estimated.
- c) Although entanglement and parameter encoding is implemented between different superconducting chips (modules), these are physically separated by only 25cm, forming a localised superconducting quantum network. As a result, these cannot be compared with distributed quantum information protocols involving, e.g. quantum memories and repeaters, capable of sharing entanglement across large distances.
- d) Despite the above negative points, I would still be tempted to support the publication of the work if it provided a significant step towards a better understanding of the superconducting platform and, e.g., the noise/imperfections that it exhibits, etc. However, the Authors, instead of proposing and verifying an accurate model for the device, empirically measure the likelihoods (probability distributions) of measurement outcomes as a function of parameters—evading the need to actually understand accurately the operation of the device.
- e) In this regard, I do not share the enthusiasm behind the maximal likelihood estimation (MLE) the Authors employ to interpret the measurement data by effectively performing tomography of the measurement-outcome probability distribution—each measurement is repeated 600 times before any parameter estimate can be even constructed. In my opinion, such an approach really disqualifies the use of the device as a sensor.

Going through the manuscript I have found the following issues, whose resolution, I hope, may still be helpful:

1. Parameter δ on the vertical axis of the lower inset of Fig 2d is undefined.
2. The title of Fig.2 “Separate metrology” and then use of “separate-measurement strategy” is confusing. I understand that the figure presents the scheme, so it should be titled something a’la “Multiparameter metrology of a remote magnetic field with...” and then in (a) what the Authors wanted to emphasise is that the scheme can be conducted in parallel, e.g. involving two pairs of qubits, rather than just the measurement is separate.
3. I find it confusing to say that the Authors design a “gradient” measurement scheme, as shown in Fig. 3, which is actually a strategy to measure difference between magnetic fields at distant sites. Moreover, they perform local estimation around the point when both field have identical vectors.
4. Fig. 3a suggests that the Bell measurement (BM) is performed simultaneously on 4 qubits, whereas these are two BMs performed on two separate qubit pairs.
5. Again, the title of Fig 3 “Directly metrology of the magnetic field gradient.” is confusing.
6. In Fig. 3 and Fig 4. “Sum of variance” → “Sum of variances”.
7. Within the Methods section it is only stated that “The vector field signal and control units are digitally simulated and encoded using U(3) formalism”. If the vector field is to represent parameters to be estimated, their actual physical interpretation should be provided already in the main text. Can the vector actually represent some external perturbation (force/field) to be sensed?

Reviewer #3

(Remarks to the Author)

The manuscript “Distributed multi-parameter quantum metrology with a superconducting quantum network” by Zhang et al.

discusses multiparameter distributed quantum sensing (DQS) in a superconducting quantum network. They demonstrate the estimation of remote vector fields and gradients.

The experiment presents some novelty with respect to other atomic or photonic experiment. In particular, I see two key aspects: 1) the parameters are encoded by generators that are locally non-commuting; 2) while in other experiment the sensing qubits or fields are physically transferred from one node to the other, here the transfer of superconducting qubit entanglement is mediated by light: as far as I understand, entanglement is generated in one module and transferred to other modules where parameter encoding happens. Maybe, another relevant aspect here are the time scales: typically, superconducting qubits involve time scales much shorter than photonic or atomic qubits. I believe these two aspects should be discussed more fully and in greater detail. An improved and more thoughtful comparison with the literature is needed (see specific comments below). I think this is the first experiment about multiparameter sensing in superconducting qubits: important differences with respect to atomic and photonic experiment are not clarified enough.

My second major criticism is the completely wrong notion of Heisenberg limit (and therefore also the claims of advantages and gain with respect to classical limits) used in this manuscript. Here N provides a sort of “amplification” of the magnetic field: that is because $U_s(x)^N = \exp(-iNB\sigma T)$ is applied, that implies $B \rightarrow NB$. So the $1/N$ estimation uncertainty is completely a classical amplification effect: it is *not* the Heisenberg limit and similarly $1/\sqrt{N}$ is *not* the classical bound (or standard quantum limit in the literature). The notion of Heisenberg limit and standard quantum limit refer to a scaling $1/N$ and $1/\sqrt{N}$, respectively, where N is the number of qubits, for multi-qubits systems. There is no notion of Heisenberg limit sensitivity for single qubits. The authors are thus using a highly misleading jargon: claiming a gain over the classical bound is just an overselling argument that does not make sense to me. This was a source of large confusion while reading. All claims of “Heisenberg limit” and “gain over classical limits” must be removed or rephrased as classical amplification effects.

The third major criticism is about the maximum likelihood analysis. I do not understand it from the description in the methods section. After acquiring a set of measurement data x_1, \dots, x_n for the given parameters y , one constructs the likelihood function $L(x_1, \dots, x_n|y) = P(x_1|y) \dots P(x_n|y)$ and maximizes it $y_{\text{est}}(x_1, \dots, x_n) = \arg\max_y L(x_1, \dots, x_n|y)$, providing the maximum likelihood estimate. This is different from the description in the manuscript. It seems that the authors do not consider actual measurement results and maximize a sort of statistical average of the likelihood function: this is different from taking the statistical average of the maximum likelihood. In other words, “the 2D landscape” of the likelihood must depend on the acquired measurement results. The unusual notion of maximum likelihood extends to the description of the control-enhanced protocol. The methods section must be rewritten to describe the true ML procedure and how it enters the adaptive protocol.

Finally, since the authors have recorded probability measurement results, I strongly suggest to compute and discuss the classical Fisher information matrix of their system, see for instance J. Phys. A 53, 023001 (2019) and arXiv:2502.17396 for reviews. The inverse of the Fisher information matrix provides the Cramer-Rao sensitivity in the system.

Let me now go in more details through the manuscript:

1) The abstract does not clearly state the novelty relative to existing work. It must explicitly position this work within multiparameter DQS and highlight concrete advances over atomic/photonic experiments.

2) The caption of figure 1 should explain what is actually shown in the figure: instead, it provides very general considerations about the system. These considerations must go in the main text and the caption should be rewritten. As far as I understand, the state transfer (orange arrows) is provided by photons: is that true? What is the difference between blue and purple regions? The authors mention “inter-module efficiency approaching 99%”: is this an original achievement of this work? Or, do they have a reference for this? Overall, a more detailed discussion of figure 1, with more details presented in the caption is needed. I think the reader needs to know more details about the transfer of entanglement mediated by light in the main text.

3) As far as I understand, the central module A in Fig. 1 is used to generate an entangled state, while the lateral modules are used for sensing. If that is true, why does the module A include sensor qubits (dark blue dots)? I am confused. What is the role of ancilla qubits? Please explain.

4) As far as I understand the first experiment involves modules A and B, while the second experiment A, B and C: this should be clarified in the last part of the “Experimental Setup” paragraph.

5) Figure 2 is somehow detached from its description in main text. To me, Fig.2a should show only module A and B, with two qubits, as discussed in the main text. This would be consistent with the other panels, which refer to the estimation of a single vectorial magnetic field, and to the discussion in the main text. At present Fig.2a is confusing. I have strong doubts about the validity of the maximum likelihood analysis shown in panel d, see discussion above.

6) In the inset of Fig. 2d, the authors show the “landscape of the likelihood function”. How do the authors do the maximum likelihood analysis if the likelihood function, as shown in the plot, has clearly a degenerate maxima? I think the reader needs more details about the maximum likelihood analysis and the adaptive protocol in the main text.

7) According to my discussion above, all claims about Heisenberg limit of xdB gain should be removed everywhere in the main text and abstract.

8) Figure 3 presents some problems as Fig. 2. Specifically, the multiple maxima of the likelihood function, the maximum likelihood analysis, as well as the claim of HL. I suggest to replace HL with $1/N$ scaling.

9) I have appreciated the comparison between local and nonlocal entanglement in Fig.4. I think this is one of the main messages of this manuscript, and real core of distributed sensing. The authors here need to acknowledge Ref. [30] and Phys. Rev. Lett. 121, 130503 (2018) that have provided a similar discussion.

10) Some overselling claims are not necessary and should be avoided. For instance, I can see the present vector field sensing applied to magnetic field sensing, but implications in “geology and astronomy” in order for “enhancing our understanding of celestial bodies and geological structures” is far from obvious and should be avoided, unless precisely explained. A discussion of time and energy scales relevant for superconducting qubits versus atomic/photonics DQS experiment is also needed (see comment above).

11) Not all the references cited in this work are relevant for the present context. At the same time, several relevant works are not cited. I think the first part of the introduction, up to “In this study ...” should be improved. The text (and the references) is very ambiguous regarding the distinction between single- and multi-parameter quantum metrology. The framework of this manuscript is the multiparameter case. This should be made clear both from the text (I suggest to clarify the multiparameter framework immediately) and from the references (a part a few relevant review on single-parameter case, I suggest to address references discussing the multiparameter case), see below.

- The sentence “Quantum metrology, which exploits quantum mechanical phenomena such as superposition and entanglement, seeks to improve measurement precision beyond classical limits.” does not refer to distributed scenario. It seems a general sentence about quantum resources for enhanced metrology: I believe reviews such as Nature photonics 5 (4), 222-229 (2011); Journal of Physics A: Mathematical and Theoretical 47, 424006 (2014); Reviews of Modern Physics 90 (3), 035005 (2018) together with major works on quantum metrology such as Physical review letters 96 (1), 010401 (2006) and Physical review letters 102 (10), 100401 (2009) should be cited instead of Refs. [1]-[7].

- Regarding distributed quantum metrology [or, more usually “Distributed quantum sensing (DQS)”] certainly Refs. [8,9,10,11,13,15] are relevant, other works are less relevant. For instance Ref. [17] is not relevant as it refers to single-parameter scenario. Alternative papers to the patent [16] by the same authors are available, such as Physical Review Research 6 (1), 013246 (2024). I also suggest reviews on multiparameter estimation, such as Physics Letters A 384 (12), 126311 (2020), and on DQS, such as arXiv:2502.17396. Other relevant publications regarding DQS are Nature communications 11 (1), 3817 (2020), which is similar to Ref. [1] but it has been published before and in a higher impact journal; Phys. Rev. A 108, 032621 (2021).

- In the present context of vector field sensing, relevant references are Physical review letters 116 (3), 030801 (2016); Nature Communications 14, 1021 (2023); npj Quantum Information 10, 98 (2024).

To summarize, certainly I cannot recommend the publication of the present version of the manuscript in Nature Communication. On the positive side, this is the first manuscript discussing distributed sensing with superconducting qubits: possible applications and interest must be clarified. On the negative side, I need to mention the misleading notion of Heisenberg limit and the unclear maximum likelihood analysis. In addition, the writing should be revised and the context of multiparameter sensing should be clarified (with discussion and relevant references). I am willing to reconsider a revised version of the manuscript, provided the authors address all my criticism.

Version 1:

Reviewer comments:

Reviewer #1

(Remarks to the Author)

The authors have thoroughly addressed my comments and critiques from the previous round of review. I am happy to recommend publication.

Reviewer #2

(Remarks to the Author)

I have carefully read the responses of the Authors to the points I have made, as well as the issues raised by the other Referees, and at this stage I am happy to recommend the article for publication in Nature Communications. As detailed below, I am satisfied and very thankful for the elaborate responses to my points, which are now resolved within the manuscript and its supplement. I still have a minor issue that should be resolved; however, as this is a presentation detail relevant to the Methods section, I will not need to inspect the manuscript again, as long as the Editor verifies this to be resolved in the next round.

In particular, the Authors have convinced me that, although parts of their protocol have already been demonstrated using the photonic platform, the current superconducting implementation constitutes a substantial and important step, including the novel gradient protocol. Moreover, even though they actually simulate the sensing protocol with pre-calibrated gates, the estimated parameters could represent physical external quantities. Furthermore, the modularity of the setup is indeed an important novelty that I have underestimated in my previous review.

I still personally think that other signal-processing techniques should be adapted to the modular superconducting-qubits

setting that go beyond the estimator construction based on maximising the likelihood (actually minimising the cross-entropy, see below), in particular, for real-time problems when sensing e.g. time-varying microwave signals. However, I agree with the Authors that these are beyond the scope of the results presented, constituting the next step for future research.

Nonetheless, I would like the Authors to still improve sections of Methods, which they have already restructured into sections “Maximum likelihood estimation for remote field sensing” and “Experimental likelihood benchmark”. I think there is some misunderstanding in the nomenclature, and this has also confused one of the other Referees, and should be unified before publishing the article.

Following a classic book such as Kay S. “Fundamentals of Statistical Signal Processing: Estimation Theory”, the MLE is constructed based only on the “model” probability distribution, i.e. for a given outcome “ i ” one maximises $\text{Log}[P_i^{\text{model}}(x,xc)]$ over x to obtain the estimate of x , which in the Author’s case would involve $P_{\text{model}}=P_{\text{ideal}}$. Now, what the Authors do is slightly different, as they maximise $L(x,xc)=\sum_i P_i^{\text{empirical}} \text{Ln}[P_i^{\text{model}}(x,xc)]$, which is more commonly referred to in the machine learning community as the minimisation of the cross-entropy loss, see e.g. (4.107-108) in Bishop, C. M. - “Pattern Recognition and Machine Learning” or Section 6.2.5 in Murphy’s “Probabilistic Machine Learning: An Introduction (MIT 2022)”. In particular, operationally the Authors by varying x minimise the Kullback-Leibler distance between $P_{\text{empirical}}$ (experiment) and P_{model} (ideal). In that sense, I encourage Authors to be a more elaborate on the estimator construction, as calling it “standard maximal likelihood estimation” is misleading.

Reviewer #3

(Remarks to the Author)

First of all, I really appreciated the new version of the manuscript: there is a huge improvement in clarity and focus compared with the first version that I reviewed months ago. The manuscript is now well focused and well written; the results and the underlying physics are very clear. The authors now rightly highlight one of the winning aspects of this work, namely the use of superconducting qubits for multiparameter distributed sensing. This platform seems crucial for transferring—via microwave photons—entanglement between distant nodes. This possibility is quite different from more standard atom/ion/photon platforms, where particles need to be physically moved from one place to another in order to realize a distributed scheme. In my opinion, this aspect fully justifies a high-impact publication. I have minor comments:

1. The authors constantly talk about “non-local entanglement,” which sounds odd to me. Yet the first word in the title is “distributed.” I thus suggest replacing “non-local entanglement” with “distributed entanglement” throughout the manuscript. For instance, the sentence “generating high-fidelity non-local entanglement” in the abstract can be replaced with “generating and distributing entanglement across a quantum network”; “To establish non-local entanglement” on page 2 can be replaced with “To distribute entanglement”; and “leverage non-local entanglement” later in the text with “exploit distributed entanglement.”

2. At the beginning of the paragraph “Sensing of remote fields,” the authors write “we simulate a setup.” What do they mean? To my understanding, this should be “we realize a setup.”

3. Regarding the references. Ref. [14], despite the title, does not discuss multiparameter estimation: it is out of context and can be removed or cited elsewhere, not together with Refs. 6–16. The list of experimental works with photons, Refs. [23–28], should include Hong et al., Nature Communications 12, 5211 (2021) and at least one work from the group of Fabio Sciarrino, e.g. Advanced Photonics 5 (1), 016005 (2023). Ref. [13] is a News & Views piece in Nature Photonics; the authors should cite the original work by Xia et al., Nature Photonics 17, 470–477 (2023), but I do not think either work is related to multiparameter sensing. Ref. [32] can be cited together with Ref. [9] for “characterization of distributed signals [6-16]”.

To conclude, I congratulate the authors on the impressive work and the revision of the manuscript. Without hesitation, I recommend publication in Nature Communications.

I. RESPONSE TO REFEREE #1

A: The authors experimentally perform two different multi-parameter quantum sensing tasks to greater than standard quantum limit precision using a network of superconducting qubit sensors:

(1) estimate all three components of a “remote” vector field located at one sensor node (i.e. they use an ancilla). In this setting it is not strictly necessary that the network be “distributed” but the motivation is a setting where one can do measurements and entangled operations in some central module and only local operations and sensing with the sensor qubit.

(2) estimate all components of a gradient of a magnetic field between two qubit sensors

The paper provides a significant experimental demonstration of a toy, yet genuinely multi-parameter, quantum sensing task using a small network of sensors and, in my opinion, could reasonably merit publication in Nature Communications provided certain things are addressed.

Reply: We sincerely thank the Referee for the positive evaluation of our work and for recognizing the significance of our experimental demonstration of multi-parameter quantum sensing using a superconducting qubit sensor network.

Some general comments:

A1: (1) It is unclear what exactly is being compared in the “sensing of remote vector fields” section when the authors claim a 6-7 dB gain for estimating the parameters. Presumably, the standard quantum limit in terms of the number of cycles N . In the abstract this is referred to as the “classical strategy” but the strategy is not classical as the sensors are fundamentally qubits. It would be more correct to refer to a “local” or “unentangled” strategy as the sensors are quantum mechanical, whether or not entanglement is used. However, it is actually unclear to me if “local” or “unentangled” is the correct characterization or if I should be comparing to using the same (entangled) states, but measuring after each cycle N ? One thing that might help is to include the standard quantum limit that is being compared to on the plots, similar to how the LE(B) and LE(O) results are compared in Fig.4e.

Reply: We thank the Referee for raising this important point. The benchmark originally referred to as the “classical strategy” represents an approach where:

1. The total sensing resource N is divided equally among the three parameters (B, θ, ϕ)
2. For each subset ($N/3$ cycles), we:
 - (i) Prepare the optimal single-qubit state for that parameter
 - (ii) Perform single-cycle evolution followed immediately by an optimal measurement
 - (iii) Repeat the procedure $N/3$ times independently

This was termed "classical individual measurement strategy" In previous studies [*Sci. Adv.*7, eabd2986 (2021), Ref. 39 in the main text].

We fully recognize the Referee's valid concern that the term "classical" could be misleading, as the sensors remain fundamentally quantum systems (qubits), and the strategy differs from using actual classical sensors by instead avoiding entanglement and collective measurements.

Changes to the manuscript: In response to the Referee's valuable suggestion, we have now replaced the phrase "*classical strategy*" with "*individual strategy*" throughout the manuscript, to indicate that this strategy estimates each parameter individually without exploiting entanglement or collective measurement. We have also explicitly labeled the SQL ($1/\sqrt{N}$) baselines, represented by the *dashed lines*, in all relevant figures and added clarifying text.

We believe these revisions make the comparisons in the plots clearer and more informative.

A2: (2) In "the distributed sensing of vector field gradients" the authors claim that without loss of generality they can take the magnetic fields B_1 and B_2 to be equal (implying that the gradient being estimated is zero) as "we can adaptively add $\nabla \vec{B}_{est}$ at site 2, which makes them asymptotically equal and does not affect the asymptotic performance. This is not obvious as the associated estimate would have to be to Heisenberg precision to make them asymptotically equal. While I expect that such an adaptive scheme is achievable it seems somewhat non-trivial to me. A reference or two and some further discussion seems necessary here.

Reply: We thank the Referee for this insightful and technically important comment. We fully agree that the feasibility of the adaptive scheme, which adds a compensating field based on a preliminary estimate $\nabla \vec{B}_{est}$, requires further clarification. The adaptive strategy uses coarse estimations to transform the estimation of the true gradient $\nabla \vec{B}$ into the estimation of the residual gradient $\nabla \vec{B} - \nabla \vec{B}_{est}$.

Figure 1: **Scaling of sum of variances under different residual gradient norms.** Main Panel: Log–log plot of the total estimation variance for the residual gradient $\nabla\vec{B} - \nabla\vec{B}_{\text{est}}$ as a function of total evolution time T . The true magnetic fields are set as $\vec{B}_1 = (\frac{\sqrt{2}}{4}, \frac{\sqrt{2}}{4}, 0)$ and $\vec{B}_2 = (0, 0, 0)$, yielding a true gradient $\nabla\vec{B} = (\frac{\sqrt{2}}{4}, \frac{\sqrt{2}}{4}, 0)$. A coarse estimate $\nabla\vec{B}_{\text{est}} = (B_{x\text{est}}, B_{y\text{est}}, 0)$ is applied at site 2, resulting in a residual gradient $\nabla\vec{B} - \nabla\vec{B}_{\text{est}} = (\epsilon_x, \epsilon_y, 0)$ to be estimated. The four curves correspond to residual norms of $\|\nabla\vec{B} - \nabla\vec{B}_{\text{est}}\| = 0$ ($\epsilon_x = \epsilon_y = 0$), $\|\nabla\vec{B} - \nabla\vec{B}_{\text{est}}\| = 0.1$ ($\epsilon_x = 0, \epsilon_y = 0.1$), $\|\nabla\vec{B} - \nabla\vec{B}_{\text{est}}\| \approx 0.36$ ($\epsilon_x = 0.2, \epsilon_y = 0.3$), and $\|\nabla\vec{B} - \nabla\vec{B}_{\text{est}}\| = 0.5$ ($\epsilon_x = 0.3, \epsilon_y = 0.4$), respectively. Inset: Scaling behavior of the total estimation variance for different orientations of the residual gradient, with the residual norm $\|\nabla\vec{B} - \nabla\vec{B}_{\text{est}}\|$ fixed at 0.5.

Here we study the distributed vector gradient sensing scheme in our experiment using a signal unitary $U_S(\nabla\vec{B} - \nabla\vec{B}_{\text{est}}, \sum \vec{B} + \nabla\vec{B}_{\text{est}})$ with non-zero residual gradient $\|\nabla\vec{B} - \nabla\vec{B}_{\text{est}}\| \neq 0$. As shown in the Figure 1, even for non-negligible offsets (e.g., $\|\nabla\vec{B} - \nabla\vec{B}_{\text{est}}\| = 0.5$ (the inset in the figure provides more details on how deviations in specific X and Y directions affect performance)), the estimation precision remains close to the ideal case for a large time interval. This shows that the adaptive control scheme is robust to moderate estimation errors. We note that the total evolution time, T , can also be adaptively adjusted. This is similar to the analysis performed in the prior work on adaptive quantum metrology [Pang et al. *Nat. Commun.* **8**, 14695 (2017), Ref. 67 in the main text].

Changes to the manuscript: These results and discussions have been included in the revised Supplementary Information (Section I.C.1) to make this point more concrete.

A3: (3) The summary section is a bit sparse and could use a little more detailed and specific discussion of implications and outlook.

Reply: We thank the Referee for the helpful suggestion. In response, we have revised the final paragraph of the *Summary* section to include a more specific discussion of the implications and future outlook. The revised paragraph is as follows:

“ These results establish a scalable framework for distributed quantum sensing of vector fields and their gradients with enhanced precision and architectural flexibility. Our approach offers a concrete path toward quantum-enhanced sensor networks that can be applied to a wide range of practical scenarios, including electromagnetic field monitoring, navigation, and remote detection. Looking forward, the integration of adaptive control, error correction, and expanded network topologies could unlock new frontiers in precision sensing and real-time quantum signal processing.”

A4: (4) Furthermore, the authors claim “through the deterministic generation of high-fidelity entanglement ...” Is 70-80% fidelity truly “high fidelity” in this context? I am a theorist so it may very well be a justified claim, but this might be a place where some more specific comparison with other systems and outlook might be helpful.

Reply: We thank the Referee for raising this important point. In our work, the reported fidelities indeed represent high levels achieved for non-local multi-qubit entanglement in superconducting architectures. Leveraging our ultra-low-loss interconnect technology, we demonstrate that both state transfer (ST) and Bell-state generation between modules reach fidelities comparable to single-chip operations, as detailed in our earlier work [Ref. 56, Niu et al. *Nat. Electron.* **6**, 267 (2023)]. In particular, we previously realized a non-local four-qubit GHZ state spanning two adjacent modules with a fidelity of 92% [Niu et al. *Nat. Electron.* **6**, 267 (2023)], which already surpasses the 86%–87% reported for on-chip four-qubit GHZ states in superconducting circuits (Barends et al., *Nature* **508**, 500–503 (2014)).

In the present work, the non-local entanglement for remote sensing strategy involves an even **more demanding setting**, where we generate a four-qubit GHZ state across two non-nearest-neighbour modules. As illustrated in Figure 2a, this protocol requires first preparing a Bell pair at the central node \mathcal{A} , distributing it via quantum state transfer to distant nodes \mathcal{B} and \mathcal{C} , and subsequently applying CNOT gates across these remote modules. This construction involves five two-qubit gates, compared to the three gates required for a conventional four-qubit GHZ state. Despite the added complexity, we obtain a fidelity of 80.36% (Figure 2b), and further transform

it into a probe state for remote sensing with a fidelity of 76.16% (Figure 2c). To our knowledge, such non-local GHZ generation across non-nearest-neighbour modules has not been previously demonstrated in superconducting systems. Given the experimental challenges, we regard these fidelities as truly “high” within this context.

Figure 2: Quantum state tomography (QST) and generation of the non-local four-qubit GHZ states. **a**, Scheme for generating a non-local four-qubit GHZ state across two non-nearest-neighbour modules. Entanglement is first created at the central node \mathcal{A} , transferred to remote nodes \mathcal{B} and \mathcal{C} , and then extended to a four-qubit GHZ state spanning \mathcal{B} and \mathcal{C} via CNOT operations. **b**, Real part of the reconstructed density matrix ρ for the non-local GHZ state across non-nearest-neighbour modules (\mathcal{B} and \mathcal{C}), with a fidelity of 80.36%. **c**, Reconstructed density matrix of the probe state generated by the non-local entanglement strategy, with a fidelity of 76.16%. **d**, Scheme for Non-local four-qubit GHZ state distributed across two adjacent modules \mathcal{A} and \mathcal{C} . **e**, Diagram and QST results for the adjacent-module case, with a fidelity of 92% [Ref.56, Niu et al., *Nat. Electron.* **6**, 267 (2023)].

For perspective, we also compare our results with other platforms. In rare-earth ion networks, entanglement fidelities across two nodes are typically in the range of 0.7–0.8 for two-qubit states, and 0.5–0.6 for three-qubit states [Ruskuc et al. *Nature* **639**, 54 (2025)]. NV center networks have achieved a two-qubit entanglement fidelity of 0.82 across two nodes, and a three-qubit GHZ state fidelity of 0.54 [Pompili et al. *Science* **372**, 259-264 (2021)]. Trapped-ion networks have reported two-qubit entanglement fidelities up to 0.86 [Main et al. *Nature* **638**, 383 (2025)]. By contrast, in the superconducting platform we realize a four-qubit non-local GHZ state with 92% fidelity across two adjacent modules, as shown in Figure 2d and Figure 2e [Niu et al. *Nat. Electron.* **6**, 267 (2023)], and, in the present work, extend this capability to non-nearest-neighbour modules with a fidelity of $\sim 80\%$.

Changes to the manuscript: We have added a sentence to the first paragraph of the “Summary” section to comment on the fidelity of the non-local entangled state and cite the relevant

papers.

“The four-qubit GHZ state between two non-nearest-neighbor nodes used in the gradient sensing exhibits fidelity of 80.36%, which is among the highest demonstrated across physically separated network nodes to date^{56,69–71} and sufficient to realize a clear advantage of the distributed protocol.”

A5: (5) Additional minor comments: In the paragraph right above “Experimental setup” on pg. 2 the authors refer to a “central modular” and a “sensor modular”. I believe this should read “central module” and “sensor module”. I.e. the system as a whole is modular, but the individual parts are modules.

Reply: We thank the Referee for catching this wording error and agree that “modular” should be replaced with “module” in this context. The terms have been corrected to *“central module”* and *“sensor module”* in the revised manuscript. We appreciate the Referee’s careful reading and helpful feedback.

II. RESPONSE TO REFEREE #2

B: The manuscript presents an experimental implementation of the sequential quantum metrology protocol introduced in [PRL 117, 160801 (2016), Ref. 58]. In short, the protocol relies on the idea that by interspersing a sequence of gates, each encoding a set of parameters, with control operations that effectively cancel the encodings, the ultimate Heisenberg limits can be reached even when dealing with multiple parameters (multiparameter metrology) generated by non-commuting Hamiltonians.

The protocol has been previously implemented on a photonic platform, first without the need to use entanglement in the single-parameter setting [PRL 123, 040501 (2019), Ref. 59], and later in the three-parameter (3D vector estimation) setting [Sci. Adv. 7, eabd2986 (2021), Ref. 57], in which entanglement between sensing and ancillary qubits (polarisation and path d.o.f.s of a single photon) was necessary to attain the multiparameter Heisenberg limit.

The current work is based on an experiment involving a different platform consisting of 5 interconnected superconducting chips (modules), each comprising 4 qubits, designed for quantum computing applications. It extends the aforementioned previous implementations of the protocol. In particular, it firstly repeats the scheme of [Sci. Adv. 7, eabd2986 (2021), Ref. 57] of sensing a 3D vector of parameters, but thanks to the usage of a modular quantum computing platform, the sensing qubit, after being entangled with an ancillary one within one chip, is transferred to another module where the encoding (and control) is performed before being transferred back again, so that a Bell measurement within the original module can be performed. The Authors then propose and implement a new version of the protocol, which they show to be suboptimal but still surpassing all classical strategies. Its task is to estimate the difference between the 3D vectors encoded at two superconducting modules, which is achieved by preparing a 4-qubit GHZ-like state involving two qubits in each of the two modules and performing Bell measurements at each module. Because both schemes involve entanglement between different modules, the Authors present them as examples of distributed quantum metrology.

Reply: We thank the Referee for carefully reading our manuscript and for providing detailed and constructive feedback, which has been invaluable in improving the clarity and quality of our work.

Although I find the achieved results very impressive, I cannot recommend the work to be published in Nature Communications for the following reasons:

Reply: We sincerely thank the Referee for his/her careful reading and thoughtful comments. We realize that certain aspects of our work may not have been clearly conveyed in the initial submission. We apologize for any confusion this may have caused. We have rewritten several parts of the manuscript to better articulate the main contributions and highlight the novel aspects of our work. These revisions aim to improve both the accessibility and the technical clarity of our work. Detailed, point-by-point responses to the Referee's comments are provided below.

B1: a) The concept and motivation behind the work has already been achieved in the aforementioned works on the photonic platform.

Reply: We thank the Referee for highlighting the prior photonic implementations and appreciate the opportunity to clarify how our work extends substantially beyond them, both conceptually and technically.

While our initial experiment is indeed inspired by the 3D vector estimation protocol demonstrated in [*Sci. Adv.* **7**, eabd2986 (2021), Ref. 57], our implementation goes beyond a mere repetition in several critical aspects. First, we realize a genuinely **distributed** quantum sensing architecture across physically separated superconducting quantum modules, connected via **low-loss microwave links**. In contrast to photonic systems, where the sensing and ancilla degrees of freedom are co-located within a single photon, our system employs high-fidelity **quantum state transfer** and **deterministic non-local entanglement distribution** between distinct nodes. This architecture allows us to separate sensing, control, and measurement physically, a key requirement for practical sensor networks to probe spatially varying signals.

Second, we introduce and experimentally demonstrate a new protocol for estimating **spatial gradients of 2D and 3D vector fields** using non-local entanglement shared across remote sensor modules. The demonstrated advantage over local-entanglement strategies captures the essential power of distributed quantum sensing. To our knowledge, such vector gradient sensing has no precedent in other systems, which have so far focused primarily on scalar parameter estimation.

Furthermore, the superconducting platform enables **real-time adaptive control, high-fidelity quantum nondemolition readout, and dynamic routing of non-local entanglement across modular hardware**. These capabilities position our system as a versatile and scalable platform

for **distributed quantum metrology**.

In summary, while our work builds upon foundational ideas from prior studies, it establishes a new regime of truly distributed, multi-parameter quantum sensing. The ability to perform vector gradient sensing and dynamically reconfigure inter-module entanglement links introduces functionalities currently beyond the reach of existing photonic platforms.

Changes to the manuscript: To better reflect these distinctions and clarify our motivation, we have revised the first paragraph of the *Introduction* section of the main text accordingly.

B2: b) The superconducting platform used is designed for quantum computing and, as I cannot really view it as sensor (network), I find the presented results more as a simulation of the quantum metrology protocol of [PRL 117, 160801 (2016), Ref. 58], which I doubt to be applicable to any real-life sensing task with external parameters to be estimated.

Reply: We appreciate the Referee's critical perspective and the opportunity to clarify the practical implications of our work. While it is correct that our experiment takes inspiration from the quantum metrology protocol in [*Phys. Rev. Lett.* **117**, 160801 (2016), Ref. 36 now], our implementation goes beyond a theoretical simulation. It represents an experimentally validated realization of a distributed quantum sensing architecture based on superconducting circuits, with demonstrated functionality and clear potential for practical extensibility.

Although superconducting platforms have primarily been developed for quantum computation, their application to quantum sensing has gained growing interest in recent years. **The architecture employed in our work leverages several unique features of superconducting qubits in a manner that is experimentally non-trivial, practically extendable, and relevant to realistic sensing tasks.**

Below, we outline these features and explain how they support the application of our protocol to real-life estimation of external physical signals.

(1) Intrinsic sensitivity to external signals

Superconducting qubits (such as transmons, flux qubits, and charge qubits) are nonlinear LC oscillators whose anharmonicity arises from embedded Josephson junctions. These junctions enable the definition of a well-isolated two-level system and simultaneously endow the qubits with strong coupling to a wide range of physical signals.

Superconducting qubits are intrinsically sensitive to:

- Microwave electromagnetic field. Qubits couple directly to microwave signals near their transition frequency via capacitive (electric field) or inductive (magnetic field) coupling. The system is described by the driven qubit Hamiltonian [Blais et al. *Rev. Mod. Phys.* **93**, 025005(2021)]

$$H(t) = \frac{\omega_q}{2}\sigma_z + \Omega \cos(\omega_d t + \phi) \sigma_x,$$

with the qubit frequency ω_q , the microwave signal frequency ω_d , amplitude (Rabi frequency) Ω and the phase ϕ . In the rotating frame of the qubit frequency

$$H_{\text{rot}} = \frac{\Delta}{2}\sigma_z + \frac{\Omega}{2}(\cos \phi \sigma_x + \sin \phi \sigma_y),$$

$\Delta = \omega_q - \omega_d$ is the detuning between qubit and signal. This Hamiltonian corresponds directly to the vector field model described in the main text, where the field components are encoded along the Pauli axes and our sensing protocol enables estimation of Δ as well as the real and imaginary parts of the complex drive amplitude $\Omega e^{i\phi}$ simultaneously. **This microwave sensing scheme can be directly implemented on our platform**, as it is inherently compatible with the system's microwave control architecture. In our experiments, **we instead simulate the signal unitaries using well-calibrated quantum gates in order to benchmark the sensing protocol under well-controlled conditions.**

- Magnetic fields, via their SQUID-based design. Many superconducting qubits incorporate a SQUID loop, allowing the qubit frequency to be tuned by an external magnetic flux. This provides high sensitivity to magnetic fields, making superconducting qubits excellent candidates for quantum magnetometry [Lachance-Quirion et al. *Science* **367**, 425-428(2020); Tabuchi, et al. *Science* **349**, 405-408 (2015); Toida et al. *Commun. Phys.* **6**,19 (2023).].
- Electric fields, charge noise, and microwave photons, all of which can be transduced into frequency shifts of the qubit. In particular, superconducting qubits are highly sensitive to ionizing radiation and cosmic ray events, which induce correlated charge bursts and quasi-particle poisoning, making them potential detectors for rare high-energy events [Li, et al. *Nat. Commun.* **16**, 4677 (2025); Fowler et al. *PRX Quantum* **5**, 040323 (2024).]. Furthermore, these qubits exhibit strong nonlinear coupling to microwave photons, enabling precise single-photon detection and quantum-enhanced microwave metrology [S. Kono, et al. *Nat. Phys.* **14**, 546 (2018); Deng et al. *Nat. Phys.* **20**, 1874 (2024); Assouly et al. *Nat. Phys.* **19**,

1418 (2023).].

- Dark matter or weak field interactions, as shown in recent superconducting qubit and cavity-based detection proposals [Dixit et al. *Phys. Rev. Lett.* **126**, 141302 (2021); Tang et al. *Phys. Rev. Lett.* **133**, 021005 (2024)].

This intrinsic responsiveness forms the physical basis of our signal model in the vector field sensing and gradient estimation experiments.

(2) Fast and High-Fidelity Control and Measurement

Superconducting qubits offer exceptionally fast and high-fidelity quantum control and quantum nondemolition (QND) measurement, both crucial for adaptive metrology and sequential estimation protocols. Single- and two-qubit gates typically operate on **nanosecond timescales** with fidelities exceeding 99.5%. State-of-the-art QND measurements can be performed at 5 ns using quantum-limited amplifiers [Ye et al. *Sci. Adv.* **10**, eado9094 (2024)] and 140 ns without amplifiers [Chen et al. *npj Quantum Inf.* **9**, 26 (2023).] with readout fidelities above 99%. Both control and measurement times are much shorter than qubit coherence times ($T_1, T_2 \sim 10\text{--}100 \mu\text{s}$), enabling real-time feedback within a single coherence window. This speed allows superconducting qubits to **sense and respond to fast-varying signal fields on sub-microsecond timescales**, a regime difficult to access with other quantum platforms.

Compared to other platforms:

Neutral atoms: Single-qubit operations typically take tens of microseconds, and entangling gates can take hundreds of microseconds to milliseconds, while QND readout times are on the order of milliseconds [Lis et al. *Phys. Rev. X* **13**, 041035 (2023)], compare to the gate time and with fidelity $> 98\%$. Though these systems achieve high fidelities, their slow operation (about 10^6 times slower than superconducting qubits) may limit measurement repetitions. However, the long-lived atomic states and exquisite sensitivity to certain fields make them excellent for applications in different scenario as superconducting qubit such as precision timing and probing fundamental physics [Ye and Zoller. *Phys. Rev. Lett.* **132**, 190001 (2024)].

NV centers: Microwave-based single-qubit gates usually take tens to hundreds of nanoseconds, and optical initialization and readout processes take microseconds. Non-destructive readout using a nuclear spin mapped to the electron spin is fidelity-limited (91%) [Bartling et al. *Phys. Rev. Applied* **23**, 034052 (2025)].

Photonic system has fast operation timescale typically determined by photon travel times. However, photonic platforms lack QND readout, with standard photodetection being destructive, which prevents repeated adaptive measurements.

Superconducting qubits thus uniquely combine **nanosecond-scale gates**, **sub-100 ns QND measurements**, and **high fidelity**, enabling favorable measurement repetition, rapid sequential feedback and adaptive control. These capabilities implemented in our experiment are highly practical relevant and directly realize and extend the theoretical framework of Ref. [58] [*Phys. Rev. Lett.* **117**, 160801 (2016)] (Methods: Adaptive control-enhanced sensing protocol and Extended Data Fig. 1).

(3) Compatibility with Quantum Error Correction (QEC)

Unlike most other sensor platforms, superconducting circuits can implement active quantum error correction schemes in real time and apply quantum control and feedback at the hardware level thanks to the QND capability [Google Quantum AI, *Nature* **638**, 920–926 (2025).; Sivak et al. *Nature* **616**, 50-55 (2023)]. The applicability of QEC is not limited to fault-tolerant quantum computing. It is also crucial for long-term scalable, robust and fault-tolerance sensing that overcome noise and decoherence [Demkowicz-Dobrzański et al. *Phys. Rev. X* **7**, 041009 (2017); Zhou et al. *Nat. Commun.* **9**, 78 (2018); Górecki et al. *Quantum* **4**, 288 (2020)]. Our platform is well-positioned to integrate QEC with quantum sensing protocols in future developments to explore robust and fault-tolerant sensor networks for multi-parameter estimation.

(4) Modularity

The architecture used in our experiment demonstrates low-loss, high-fidelity state transfer and generation of non-local entanglement across modules [Ref. 56, Niu et al. *Nat. Electron.* **6**, 267 (2023)]. It enables distributed quantum protocols including parallel operation across sensors, spatially separated sensing and gradient estimation, which are key for practical metrology applications such as magnetic gradiometry or field mapping.

Our modular protocol is more than a simulation — it is an experimentally validated implementation of a real distributed sensing strategy. This point is elaborated further in the reply to the next comment.

Changes to the manuscript: We have made corresponding revisions in the *Introduction* section to point out the key features of superconducting qubit platform as quantum sensors and cited relevant applications. Please refer to the second paragraph of the *Introduction* for details, where the text now reads: “*Superconducting circuits offer a powerful platform for DQM, ... position-*

ing this platform to implement advanced DQM protocols with real-time feedback and dynamic reconfigurability.”

B3: c) Although entanglement and parameter encoding is implemented between different superconducting chips (modules), these are physically separated by only 25 cm, forming a localised superconducting quantum network. As a result, these cannot be compared with distributed quantum information protocols involving, e.g. quantum memories and repeaters, capable of sharing entanglement across large distances.

Reply: We thank the referee for the thoughtful comment regarding the spatial separation of superconducting modules in our architecture. While the physical distance between modules in our current experiment is 25 cm, this does not reflect any fundamental limitation of the architecture. Importantly, our platform realizes a **genuinely nonlocal modular quantum network**, where independently fabricated, packaged, and controlled superconducting chips are spatially separated and coherently entangled through engineered low-loss interconnects and tunable couplers. This represents true distributed quantum protocols in the following important senses.

(1) Genuine modular structure and inter-module entanglement

The significance of our work lies not in the absolute spatial separation but in the successful realization of **distributed quantum protocols**—including remote entanglement generation, nonlocal parameter encoding, and multi-node multi-parameter quantum sensing—across independently controlled quantum modules. This fundamentally distinguishes our approach from monolithic superconducting platforms where all qubits reside on a single chip under global control.

Our architecture incorporates the essential ingredients for scalable modular quantum networks. Built on a high-coherence superconducting platform, it achieves Bell-state fidelities up to 99% between physically distinct chips, as demonstrated in our prior work [Ref. 56, Niu et al. *Nat. Electron.* **6**, 267 (2023)]. The protocols generate nonlocal multi-qubit entanglement across independently fabricated, packaged, and controlled modules. Each module can be selectively exposed to external fields for sensing while preserving quantum coherence and inter-module entanglement, thereby enabling genuine distributed sensing.

(2) Extensibility to long-range quantum networks

The foundational principles of distributed quantum sensing—modular entanglement, nonlocal encoding, and cooperative multi-node protocols—are scale-invariant with respect to network size.

While extending to long-distance quantum networks will require photonic links, quantum memories, and repeaters to overcome loss and decoherence, the protocol structure and network paradigm demonstrated here are fully extensible to such regimes.

Recent progress in superconducting quantum systems has already demonstrated entanglement distribution over tens of meters [Storz et al., *Nature* **617**, 265 (2023); Qiu et al., *Sci. Bull.* **70**, 351 (2025)], and hybrid superconducting–optical interfaces are actively being developed [Mirhosseini et al., *Nature* **588**, 599 (2020)]. Our modular platform therefore serves as a high-coherence testbed for distributed quantum protocols that can naturally evolve into longer-range, heterogeneous quantum networks.

(3) Practical utility of short-range distributed sensor networks

Even at centimeter-scale separations, distributed quantum sensor networks offer unique and practical advantages:

- **Remote sensing protocol:** Physical separation between sensing qubits and ancillary/control qubits allows selective exposure of certain modules to external fields while shielding others, which is a critical capability in many metrology applications.
- **Cosmic ray detection:** Cosmic rays typically affect an entire superconducting chip/module. Spatially isolated, entangled modules can perform differential sensing, and thus enables precision metrology and error-resilient distributed information processing [Wu, et al. *arXiv:2505.15919* (2025); Xu et al. *Phys. Rev. Lett.* **129**, 240502 (2022)].
- **Condensed matter magnetometry:** Distributed qubits positioned across a sample surface can map local magnetic field gradients with high spatial resolution.
- **Microwave field sensing:** For microwave signals at 10 GHz, the wavelength (~ 3 cm) is comparable to the physical separation of our modules (~ 25 cm), placing the network within the near-field or intermediate regime of typical sources. Our protocol enables spatial reconstruction of local microwave field structure and directionality.

These examples demonstrate that short-range distributed quantum networks are not only meaningful but **necessary** for a variety of sensing applications.

(4) Scalable platform for distributed quantum sensing

Our modular architecture provides a solid foundation for distributed quantum sensing, as validated by high-fidelity non-local entangled state, multi-parameter estimation, and adaptive inter-module control. The current spatial separation is limited primarily by cryogenic infrastructure, not by fundamental physical constraints.

The techniques developed here such as low-loss quantum interconnects, precise synchronization, and inter-module entanglement generation are readily transferable to multi-node and multi-cryostat systems. They are also compatible with long-range photonic interconnects and quantum repeaters architecture, offering a path toward large-scale quantum networks.

Looking forward, we aim to extend this platform beyond a single dilution refrigerator. This will involve overcoming technical challenges such as coherent quantum state transfer across temperature gradients (e.g., from 10 mK to 1 K), where thermal noise introduces substantial decoherence. Developing high-fidelity interconnects that bridge such thermal zones is a central goal of our ongoing efforts.

In summary, our experiment demonstrates a modular superconducting quantum network that achieves high-fidelity inter-module entanglement and distributed sensing across independently controlled nodes. Although currently realized at centimeter-scale separations, the architecture and protocols are practically relevant and inherently extensible to larger spatial networks, representing a significant step toward scalable, reconfigurable platforms for distributed quantum metrology.

To maintain clarity in presenting the quantum metrology protocol and in consideration of space constraints, we provided only a succinct description of the experimental architecture in the initial manuscript. We regret this omission and are grateful to the referee for offering the opportunity to clarify this essential aspect.

B4: d) Despite the above negative points, I would still be tempted to support the publication of the work if it provided a significant step towards a better understanding of the superconducting platform and, e.g., the noise/imperfections that it exhibits, etc. However, the Authors, instead of proposing and verifying an accurate model for the device, empirically measure the likelihoods (probability distributions) of measurement outcomes as a function of parameters—evading the need to actually understand accurately the operation of the device.

Reply: We sincerely thank the Referee for recognizing that our work has the potential to contribute to the understanding of the superconducting platform, including its intrinsic noise and

imperfections.

To maintain clarity in presenting the quantum metrology protocol and in consideration of space constraints, our initial manuscript included only a concise description of the experimental architecture, while more detailed discussions of device parameters, noise channels, and their experimental manifestations were presented in the Supplementary Information (Sec. II.A–C). For example, in Sec. II.B of the Supplementary Information we explicitly quantified the contributions of control errors and decoherence: *“The error in this sequence primarily stems from the control error when synchronously transferring two entangled states. The control error is estimated to be 11.44% for generating the non-local GHZ state; the decoherence error throughout this 340 ns sequence is approximately 8.34%. These two parts yield an estimated fidelity of 80.22%, which is close to our experimental result of 80.36%. The non-local entangled state across two chips, which are not directly connected, is more fragile to environmental noise. The effective decoherence rate of the probe state is estimated to be $80 \times 2\pi$ kHz. As a consequence, the fidelity values of $|\Psi_f\rangle$ are 69.27%, 63.81%, 58.78%, 54.15% for $N = 1 \sim 4$, respectively. These values are higher than the confidence threshold $\sim 50\%$ for entanglement.”* We now realize that the lack of explicit cross-referencing and contextual discussion in the main text may have hindered the overall clarity and readability. We are grateful to the Referee for providing us the opportunity to clarify and strengthen this aspect.

In addition, during our experiments, we observed that when two qubits interact at different frequencies, their joint operations accumulate an additional U(1) phase in the circuit. Achieving high-fidelity performance, therefore, requires both careful calibration of multi-qubit gates and precise phase control. In the revised Supplementary Information, we have added a new section, Sec. II.C, titled **“Gate set calibration,”** which explicitly presents **the calibration protocols for CZ gates, CNOT gates, and inter-module state-transfer operations**, together with the associated phase calibration procedures. *The detailed calibration process is provided in Sec. II.C and illustrated in Figure 3 (Fig. S5).* These additions are intended to make the experimental processes, error sources, and mitigation strategies transparent to the reader.

Changes to the manuscript: In response, we have significantly revised the manuscript to address these concerns:

- The **Introduction** has been expanded to more clearly position our work within the context of modular quantum architectures, and to emphasize the unique advantages of superconducting qubits for quantum sensing.

Figure 3: **Repeated gate sequences (drawn by Qiskit) for $U(1)$ phase calibration.** **a**, CZ phase calibration sequence. The virtual Z phase of the control qubit is fixed, while that of the target qubit is swept, and the resulting probability on the target qubit (P_t) is measured to extract the accumulated phase. **b**, CNOT phase calibration sequence. A CNOT gate is compiled into a $-Y/2$ gate, followed by a CZ gate and a $Y/2$ gate. The control qubit is initialized in the $|+\rangle$ state by a Hadamard gate, and repeated CNOT operations reveal oscillations between $\frac{1}{\sqrt{2}}(|00\rangle + |11\rangle)$ and $|00\rangle$, with the correct compensating phase maximizing the exchange, exhibited by the probability on state $|11\rangle$ (P_{11}). **c**, Calibration sequence for inter-module quantum state transfer. We transfer a $|0\rangle + i|1\rangle$ state from the transmission qubit (q_0) to the receiving qubit (q_1), and then use another state transfer to recover it back to the transmission qubit. This transfer and recover set is repeated by multiple layers, and the result is measured on X-axis of q_1 . **d–f**, Extraction of virtual Z phases from repeated sequences of CZ, CNOT gates, and state transfer operations, respectively.

- The **Experimental setup** section has been revised to include additional technical details about the superconducting platform, including microwave interconnects, inter-module gate fidelities, and the mechanism used to generate non-local entanglement across modules.
- **Gate calibration and phase compensation:** A new subsection and figure (Sec. II.C, Fig. S5) have been added to the Supplementary Information, detailing the calibration of entangling gates and the accumulation and correction of $U(1)$ phases.
- Throughout the manuscript, we have improved cross-referencing to the Supplementary Information, ensuring that the reader can easily access relevant discussions on system-level parameters and experimental imperfections.

B5: e) In this regard, I do not share the enthusiasm behind the maximal likelihood estimation (MLE) the Authors employ to interpret the measurement data by effectively performing tomography of the measurement-outcome probability distribution—each measurement is repeated 600 times before any parameter estimate can be even constructed. In my opinion, such an approach really disqualifies the use of the device as a sensor.

Reply: We thank the Referee for raising these concerns and appreciate the opportunity to clarify the role and necessity of maximum likelihood estimation (MLE) and empirical likelihood calibration in quantum sensing, particularly in the multi-parameter and multi-qubit regimes.

(1) Repeated projective measurement is a standard and necessary procedure in qubit-based quantum sensing

We appreciate the Referee’s concern about the measurement repetition and believe it may arise from a comparison with ensemble-based sensing platforms—such as atomic vapor cells or NV center ensembles—where a single measurement of a macroscopic observable (e.g., optical absorption or fluorescence intensity) effectively averages over a large number of identical particles. In such systems, one measurement often provides a continuous signal proportional to the expectation value of the observable ($\langle\sigma_z\rangle$ for example), with integration times typically on the order of milliseconds to seconds.

In contrast, superconducting qubits—and more generally, all qubit-based quantum processors—operate fundamentally via projective measurement. For example, a Z -basis measurement collapses the qubit to either $|0\rangle$ or $|1\rangle$ state with respective probabilities P_0 and P_1 . Each projective measurement yields only a single binary outcome, and repeated measurements under identical conditions are required to reconstruct the underlying probabilities. In our experiment, we perform 600 repetitions per setting to estimate probabilities and derive maximum likelihood estimators (MLEs) of the signal parameters. Importantly, the duration of each repetition is short and only on the order of microseconds, so the total measurement time remains efficient and comparable to ensemble platforms in terms of precision per unit time. This repetition-based approach is standard across all qubit-based quantum sensing and computation platforms, including superconducting, trapped-ion, and atom system. Furthermore, it is different from quantum state tomography, which would require measurements in all Pauli bases to reconstruct the full density matrix. In our case, we only extract the state populations in one fixed basis, which is sufficient for metrological inference.

Lastly, the number of repetitions is a fundamental resource in quantum metrology. It directly

determines the statistical confidence of the estimates and is explicitly accounted for in our precision analysis.

(2) MLE is essential for multi-parameter quantum sensing.

Quantum sensing involves recovering unknown signal parameters from observed probabilities. In single-parameter cases, this mapping may be analytically invertible. For example, in Ramsey spectroscopy, a qubit initially in $|+\rangle = \frac{1}{\sqrt{2}}(|0\rangle + |1\rangle)$ evolves under

$$H = \frac{\omega}{2}\sigma_z,$$

for time t yields a state $|\psi(\omega, t)\rangle = \cos(\omega t/2)|0\rangle + i \sin(\omega t/2)|1\rangle$. Measuring in the $|\pm\rangle$ basis gives a probability

$$P_+(\omega) = \frac{1}{2}[1 + \cos(\omega t)].$$

Inverting this gives a direct estimate of ω from the observed measurement statistics.

However, when estimating multiple parameters, especially those generated by non-commuting observables, the measurement probabilities depend nonlinearly and jointly on all parameters. For instance, consider evolution under

$$H = B_x\sigma_x + B_z\sigma_z$$

with the qubit initially in $|0\rangle$. The final state and measurement statistics now depend jointly on both B_x and B_z , and their effect cannot be separated cleanly. For example, the measured probability of $|1\rangle$ state takes a form

$$P_1(B_x, B_z) = \frac{B_x^2 \sin^2(\sqrt{B_x^2 + B_z^2} t)}{B_x^2 + B_z^2}.$$

In such cases, there is no analytic inversion to recover (B_x, B_z) from a single measurement outcome. Instead, one must collect measurement data and use statistical inference, typically MLE or Bayesian methods, to estimate the parameters. This practice is well-established in quantum metrology [e.g. Giovannetti et al. *Nat. Phys.* **5**, 222 (2011); Valeri et al. *npj Quantum Inf.* **6**, 92 (2020); Marciniak et al. *Nature* **603**, 604 (2022)], where likelihood-based estimators are the standard tool for multi-parameter quantum estimation.

(3) MLE is not a workaround but a provably optimal method.

MLE is not an ad hoc or heuristic technique. Under standard conditions, it is: (a) Asymptotically unbiased [Wasserman, L. *All of Statistics: A Concise Course in Statistical Inference* (Springer, 2004)]; (b) Statistically efficient — it saturates the Cramér–Rao bound and, for optimal

measurements, the quantum Cramér–Rao bound [Paris, *Int. J. Quant. Inf.* **7**, 125 (2009)]; (c) Robust to experimental noise, including control errors and SPAM noise [Granade et al. *New J. Phys.* **14** 103013 (2012); Zhao et al. *Phys. Rev. Research* **5**, 023136 (2023); Hou et al. *Sci. Adv.* **7**, eabd2986 (2021) and Supplementary Information (Sec.III)].

These properties make MLE the method of choice for multi-parameter estimation in both classical and quantum sensing. Our numerical simulations and experimental results validate its accuracy and convergence in the presence of realistic noise.

(4) Empirical likelihoods reflect the physical behavior of real devices.

Rather than assuming idealized models, we empirically map the likelihood landscape through calibration data. This captures realistic effects such as gate errors, readout nonidealities, cross-talk, and decoherence, which are difficult to model analytically but are critical to the sensor’s actual performance.

As discussed in Sec. II and III of the Supplementary Information, we benchmark device performance using these probabilities and empirical likelihoods measured from experiment. We extract the MLE landscape, study its structure and compare it to numerical models. This landscape captures how the control settings and underlying signal affect the inferred parameters. Similar techniques have become standard in quantum sensing, especially those using statistical inference for parameter estimation [Valeri et al. *npj Quantum Inf.* **6**, 92 (2020); Bonato et al. *Nat. Nanotechnol.* **11**, 247 (2016); Marciniak et al. *Nature* **603**, 604 (2022)].

Our approach consists of two steps: (i) benchmarking the sensor by constructing the MLE likelihood landscape, and (ii) estimating unknown parameters from experimental data. When prior knowledge of the signal parameters is available, control parameters are chosen to optimize sensitivity. Otherwise, adaptive feedback is used to approach the optimal regime dynamically. This two-step process yields both diagnostic insight and practical estimation with high precision.

Importantly, this strategy is also future-proof: in many-body quantum sensors or interacting quantum networks, the sensing dynamics may be chaotic or analytically unsolvable. In such cases, empirical benchmark and estimation is essential. Our methodology is thus aligned with the requirements of next-generation sensing platforms [e.g., Marciniak, et al. *Nature* **603**, 604 (2022)].

(5) Measurement repetitions are standard and do not disqualify sensing.

Repeating quantum measurements to construct a probability distribution is fundamental to all quantum platforms because individual measurements are inherently probabilistic. Our choice of

600 repetitions per setting (600 repeating measurements to get one set of state probabilities, and thus one estimation of the signal parameters) balances precision and experimental overhead. With fast readout and reset, this corresponds to a few seconds of runtime and is less than one second in our next generation devices, which is largely limited by the speed of classical signal processing. The quantum measurement and reset only takes $\sim 10^{-4}$ s if start-of-the-art technology in superconducting circuit is used. This overhead could be further reduced by improving FPGA device to speedup classical processing, and using more sophisticated estimation protocol that involve adaptive measurement allocation or real-time Bayesian updating to lower the required number of repetition. Compared to other platforms (e.g., NV centers, cold atoms ensembles), superconducting circuits offer a favorable repetition rate, fast control and measurement.

(6) Our platform integrates MLE with adaptive control.

In our experiment, MLE is not used solely for post-processing. It is embedded within an adaptive feedback loop to dynamically update control settings in real-time, enabling efficient navigation of the parameter space. Adaptive likelihood-based strategies like this have proven effective in recent experimental metrology studies [Marciniak et al. *Nature* **603**, 604–609 (2022)].

In summary, MLE and empirical likelihood calibration are not limitations of our approach — they are necessary and powerful tools for performing multi-parameter quantum sensing in realistic platforms. Our method builds on the physical behavior of the device and enables robust and scalable estimation protocols, which are essential for both current and future quantum sensors based on superconducting hardware. We respectfully disagree with the assertion that this approach disqualifies our platform as a sensor and believe it demonstrates a viable and powerful sensing architecture.

Changes to the manuscript: To address this comment, we (i) revised *Methods* to show explicitly that our maximum likelihood estimator is constructed from measured outcome counts in a fixed basis; (ii) separated the procedures into two subsections—“*Maximum likelihood estimation for remote field sensing*” (used for estimation) and “*Experimental likelihood benchmark*” (diagnostic only, not used for estimation/feedback).

B6: Going through the manuscript I have found the following issues, whose resolution, I hope, may still be helpful:

1. Parameter delta on the vertical axis of the lower inset of Fig 2d is undefined.

Reply: We thank the Referee for carefully reading our manuscript and pointing out this. The parameter δ on the vertical axis of the lower inset in Fig.2d represents the standard deviation, which quantifies the assessed precision of the estimated parameters. To avoid ambiguity, we have now explicitly clarified this in the revised figure caption, which reads: “...*Lower inset: Assessed precision (standard deviation δ) of the three parameters...*”.

B7: 2. The title of Fig.2 “Separate metrology” and then use of “separate-measurement strategy” is confusing. I understand that the figure presents the scheme, so it should be titled something a’la “Multiparameter metrology of a remote magnetic field with...” and then in (a) what the Authors wanted to emphasise is that the scheme can be in conducted in parallel, e.g. involving two pairs of qubits, rather than just the measurement is separate.

Reply: We thank the Referee for this helpful suggestion. We agree that the original title “Separate metrology” may be misleading and does not accurately capture the intention or structure of the figure. The purpose of Fig.2 was to present an alternative configuration of the gradient estimation protocol employed in our second experiment, where entangled resources are distributed from Module \mathcal{A} to both Modules \mathcal{B} and \mathcal{C} . In this configuration, local magnetic fields at Modules \mathcal{B} and \mathcal{C} are separately estimated using a sensor–ancilla network, thereby calculating the gradients. This configuration and the associated precision analysis are described in detail in the Supplementary Information Sec.I.C.

Changes to the manuscript: Following the Referee’s recommendation, we have revised the title of Fig.2 to “*Multiparameter quantum metrology of a remote magnetic field with the sensor-ancilla network.*”, and updated Fig.2a accordingly to better reflect the scenario discussed in the main text. To further avoid potential ambiguity, the original schematic shown in Fig.2a has been revised and moved to the **Supplementary Information (Fig.S2)**, where we now clearly distinguish the experimental configurations in both the figure caption and accompanying text.

B8: 3. I find it confusing to say that the Authors design a “gradient” measurement scheme, as shown in Fig. 3, which is actually a strategy to measure difference between magnetic fields at distant sites. Moreover, they perform local estimation around the point when both field have identical vectors.

Reply: We appreciate the Referee’s insightful comment. We fully agree with the interpretation

that our scheme measures the difference between magnetic fields at spatially separated sensor nodes. The term “gradient” is used in our manuscript to denote a finite-difference estimate of a spatial derivative, a standard approach when probing continuous fields using discrete sampling points. In this context, the field difference naturally serves as an estimator for the gradient.

Regarding the assumption that the two magnetic fields are locally identical, we emphasize that **this condition is not an a priori requirement but can be effectively realized through an adaptive control protocol**. When the gradient is nonzero, we can add a compensating local field at one sensor node according to a preliminary estimate of the gradient $\nabla\vec{B}_{\text{est}}$. This transforms the original estimation problem into one where the residual gradient $\nabla\vec{B} - \nabla\vec{B}_{\text{est}}$ becomes the new estimation target. Importantly, the adaptive protocol does not require $\nabla\vec{B}_{\text{est}}$ to be highly precise from the outset.

To clarify the feasibility of the proposed adaptive scheme, we numerically analyze how the estimation precision behaves under varying levels of residual gradient $\nabla\vec{B} - \nabla\vec{B}_{\text{est}}$. As shown in Figure 1, even for sizable deviations (e.g., $\|\nabla\vec{B} - \nabla\vec{B}_{\text{est}}\| = 0.5$), the total estimation variance remains close to the ideal case over a sufficiently long evolution time T . The inset further illustrates how deviations along the X and Y directions individually affect the estimation performance. This robustness underscores the feasibility of implementing the scheme without precise initial knowledge. We note that the total evolution time, T , can also be adaptively adjusted. Such iterative refinement techniques are well-established in quantum metrology (e.g., Pang et al. *Nat. Commun.* **8**, 14695 (2017), Ref. 67 in the main text) and ensure that the assumption of “identical vectors” can be effectively realized in practice, without requiring an exact match from the outset.

Changes to the manuscript: These clarifications, together with the supporting numerical results, have been included in **Section I.C.1 of the revised Supplementary Information**.

B9: 4. Fig. 3a suggests that the Bell measurement (BM) is performed simultaneously on 4 qubits, whereas these are two BMs performed on two separate qubit pairs.

Reply: We thank the Referee for the careful observation. To accurately reflect the protocol, which involves two independent Bell measurements on separate qubit pairs, we have **revised Fig.3a by replacing the original single BM block with two clearly separated BM blocks**.

B10: 5. Again, the title of Fig 3 “Directly metrology of the magnetic field gradient.” is confusing.

Reply: We thank the Referee for this comment. The original title “Direct metrology” was intended to contrast the sensing strategy in Fig.3 with the “separate” metrology approach shown in the original Fig.2a and in Fig.4b. In the direct scheme, a non-local entangled probe state is used such that information about the magnetic field gradient is encoded in the final state. This allows the gradient to be directly extracted from the measurement probabilities, without the need to separately estimate the local magnetic fields. In contrast, the separate scheme involves individual estimation of the local fields at each site, followed by classical postprocessing to compute the gradient.

Changes to the manuscript: To improve clarity and eliminate potential ambiguity, we have revised the title of Fig.3 to “*Multi-parameter quantum metrology of the magnetic field gradient*”. In addition, a detailed comparison between the direct and separate metrology strategies has been included in the Supplementary Information (Sec. I.C.3) for clarity.

B11: 6. In Fig. 3 and Fig 4. “Sum of variance” → “Sum of variances”.

Reply: We thank the referee for spotting this. We have corrected “*Sum of variance*” to “*Sum of variances*” in both Fig. 3 and Fig. 4, as well as the Supplementary Fig.S14 and Fig.S15.

B12: 7. Within the Methods section it is only stated that “The vector field signal and control units are digitally simulated and encoded using U(3) formalism”. If the vector field is to represent parameters to be estimated, their actual physical interpretation should be provided already in the main text. Can the vector actually represent some external perturbation (force/field) to be sensed?

Reply: We thank the Referee for highlighting the need for a clearer physical interpretation of the vector field signal in the main text.

As discussed in Sec. II B of the Supplementary Information, the signal and control operations in our experiment are physically implemented using the gate decomposition

$$U_3(\alpha, \beta, \lambda) = R_z(\beta)R_x\left(\frac{\pi}{2}\right)R_z(\alpha)R_x\left(-\frac{\pi}{2}\right)R_z(\lambda),$$

where R_x and R_z are single-qubit rotations realized by phase- and amplitude-modulated microwave pulses. These pulses are generated via an arbitrary waveform generator and delivered to the qubits through dedicated microwave lines (see also Sec. II A of the Supplementary Information).

This unitary-based encoding simulates the action of a real physical signal without loss of gener-

ality. In realistic sensing scenarios, superconducting qubits can couple to a wide range of external signals such as microwave field, magnetic field, or charge noise, as discussed in the reply to the second comment (B2), via interactions of the form

$$U_s(\hat{x}) = \exp(-i\vec{B} \cdot \sigma T),$$

where \vec{B} encodes the parameters of the external perturbation to be estimated (e.g., field amplitude, direction, or phase), and σ denotes the vector of Pauli operators acting on the qubit.

Rather than applying a physical perturbation directly, we digitally encode its effect through calibrated gate sequences. This allows us to systematically benchmark the performance of our multi-parameter estimation protocol across arbitrary directions in parameter space, while remaining fully general with respect to the physical origin of the signal. Moreover, this strategy is well aligned with other quantum sensing and Hamiltonian estimation protocols using simulated Hamiltonian evolution [e.g., Pastori et al. *PRX Quantum* **3**, 3 030324 (2022); Wang et al. *Nat. Phys.* **13**, 551 (2017)].

Changes to the manuscript: We have clarified this point in the revised main text and Methods section. In the first paragraph of section “*Sensing of remote vector fields*”, we have added “*In our experiment, the signal unitary is implemented using a sequence of calibrated quantum gates to accurately emulate field-induced quantum evolution. This approach enables precise and controlled benchmarking of the sensing protocol under well-defined conditions. For practical applications such as microwave sensing, the vector field components may represent physical parameters including field amplitude, phase, and frequency detuning.*”

In the “*Experimental platform*” subsection of the Methods, we have added relevant descriptions and explicit references to the corresponding experimental details provided in the Supplementary Information. Specifically, we now state: “*The vector field signal and control unitaries are digitally simulated using a $U(3)$ formalism (see Supplementary Information, Section II.B), where the signal parameters are set to span representative conditions, while control parameters are set according to prior experimental calibration. To extract information about the encoded signals parameters, Bell measurements are performed on selected qubits to obtain the output state probabilities.*”

III. RESPONSE TO REFEREE #3

C0: The manuscript “Distributed multi-parameter quantum metrology with a superconducting quantum network” by Zhang et al. discusses multiparameter distributed quantum sensing (DQS) in a superconducting quantum network. They demonstrate the estimation of remote vector fields and gradients.

The experiment presents some novelty with respect to other atomic or photonic experiment. In particular, I see two key aspects: 1) the parameters are encoded by generators that are locally non-commuting; 2) while in other experiment the sensing qubits or fields are physically transferred from one node to the other, here the transfer of superconducting qubit entanglement is mediated by light: as far as I understand, entanglement is generated in one module and transferred to other modules where parameter encoding happens. Maybe, another relevant aspect here are the time scales: typically, superconducting qubits involve time scales much shorter than photonic or atomic qubits. I believe these two aspects should be discussed more fully and in greater detail.

Reply: We sincerely thank the Referee for the thoughtful evaluation and for recognizing the novelty of our approach to distributed quantum metrology using superconducting quantum networks. We greatly appreciate the Referee’s insightful identification of the key aspects that distinguish our experiment from prior works in atomic and photonic systems. In response, we have revised the manuscript to clarify and expand on these critical points, which we summarize and elaborate upon below.

(1) Non-commuting generators in multiparameter encoding

The Referee rightly highlights that one novel aspect of our work lies in the fact that the parameters are encoded via locally non-commuting generators—a scenario that is fundamentally more complex and richer than the commonly studied case of commuting observables. In the revised manuscript, we have discussed this feature in greater detail in the introduction, stating: *“a critical open challenge for practical distributed quantum metrology is the extension to multi-parameter sensing with non-commuting generators. In this more complex scenario, standard estimation strategies are highly inefficient^{6,30–33}. Successfully overcoming this barrier will require two key advancements: first, the generation of high-quality, genuine nonlocal entanglement between remote nodes, and second, the design of metrological protocols capable of simultaneously estimating multiple parameters with high precision^{34–39}.”* Furthermore, in the main text, we explicitly connect

this challenge to our implementation of non-local entangled state and control-enhanced sequential estimation strategies, which are essential to achieving performance beyond standard quantum limits in the presence of generator incompatibility.

(2) Physical architecture and entanglement distribution

We agree with the Referee that our entanglement distribution method represents a significant departure from existing architectures in which either the qubits or the mediating photons are physically transported between nodes. As the Referee noted, our architecture does not physically move qubits or signals across nodes, but instead leverages high-fidelity quantum state transfer through microwave resonators to distribute entanglement between modules. This mode of non-local entanglement generation allows flexible allocation of resources between sensing and control, and is highly scalable.

To clarify this point, we have revised the description of our experimental setup as follows: *“It comprises multiple sensor modules connected to a central module \mathcal{A} via four 25-cm aluminum coaxial cables, which serve as low-loss transmission lines for microwave photons. Each module hosts four transmon qubits, and the interface between the qubit and cable is equipped with a tunable coupler that enables programmable interaction between qubits and cable modes. The cables function as multimode resonator buses that support standing wave modes, enabling the coherent transfer of microwave photons between modules. By carefully coordinating controls on the qubits and couplers, the system can achieve high-fidelity inter-module operations, with state transfer efficiencies approaching 99%. To establish non-local entanglement between modules, entangled states are first prepared locally within the central module. The quantum state of one or more qubits is then coherently transferred to remote modules.”*

It should be noted that this indirect yet coherent transfer of entanglement via resonator-mediated interactions is both modular and scalable and provides a versatile platform for implementing distributed sensing protocols without physically transporting matter-based qubits.

(3) Fast timescales and microwave compatibility of superconducting qubits

As the Referee noted, the ultrafast operational timescales of superconducting qubits (nanoseconds to microseconds) distinguish them from atomic or photonic systems and pose both technical challenges and metrological opportunities. This characteristic imposes stringent demands on entanglement distribution, control synchronization, and readout fidelity (see also the reply to Referee B’s comment B2).

In the revised manuscript, We have expanded the introduction to highlight these platform-

specific advantages: *“Superconducting circuits offer a powerful platform for DQM, uniquely combining high-speed quantum operations with scalable networking capabilities. Their nanosecond-scale gate times and native compatibility with microwave control and signal transduction^{40–43} make them particularly well-suited for detecting fast, weak signals, such as those arising in dark matter and cosmic ray detections^{44–47}.”*

We hope that these clarifications and revisions address the Referee’s insightful concerns and further underscore the conceptual and technical significance of our work in advancing the field of distributed multiparameter quantum metrology.

C1: An improved and more thoughtful comparison with the literature is needed (see specific comments below). I think this is the first experiment about multiparameter sensing in superconducting qubits: important differences with respect to atomic and photonic experiment are not clarified enough.

Reply: We thank the Referee for emphasizing the importance of situating our work within the broader context of quantum metrology and for recognizing its novelty as, to our knowledge, the first experimental demonstration of multiparameter sensing using superconducting qubits. We fully agree that a clearer comparison with prior studies in atomic and photonic platforms is essential to highlight the distinct contributions of our work.

In response to this constructive suggestion, we have revised the introduction to explicitly contrast our work with existing experiments in distributed quantum sensing and cited relevant papers. Specifically, we now state: *“Recent demonstrations using photonic^{23–28} and atomic²⁹ platforms have proven that entanglement distributed across nodes can significantly enhance single-parameter sensing. These implementations typically measure global properties, such as an average phase, encoded by mutually commuting generators. Despite the progress, a critical open challenge for practical distributed quantum metrology is the extension to multi-parameter sensing with non-commuting generators. In this more complex scenario, standard estimation strategies are highly inefficient^{6,30–33}. Successfully overcoming this barrier will require two key advancements: first, the generation of high-quality, genuine nonlocal entanglement between remote nodes, and second, the design of metrological protocols capable of simultaneously estimating multiple parameters with high precision^{34–39}.”*

We further clarify that superconducting qubits offer unique advantages for addressing this chal-

lenge (see response to Comment C0.(3)), including fast, high-fidelity gate operations on nanosecond timescales, adaptive control, and deterministic generation and routing of non-local entanglement across modular quantum hardware. These capabilities enable more flexible and scalable approaches to distributed multiparameter sensing, which are difficult to realize with atomic or photonic platforms.

We hope that these clarifications more effectively contextualize our contribution and highlight the distinctive advantages of superconducting circuits in advancing distributed quantum metrology.

C2: My second major criticism is the completely wrong notion of Heisenberg limit (and therefore also the claims of advantages and gain with respect to classical limits) used in this manuscript. N here provides a sort of “amplification” of the magnetic field: that is because $U_S(x)^N = \exp(-INB\sigma T)$ is applied, that implies $B \rightarrow NB$. So the $1/N$ estimation uncertainty is completely a classical amplification effect: it is *not* the Heisenberg limit and similarly $1/\sqrt{N}$ is *not* the classical bound (or standard quantum limit in the literature). The notion of Heisenberg limit and standard quantum limit refer to a scaling $1/N$ and $1/\sqrt{N}$, respectively, where N is the number of qubits, for multi-qubits systems. There is no notion of Heisenberg limit sensitivity for single qubits. The authors are thus using a highly misleading jargon: claiming a gain over the classical bound is just an overselling argument that does not make sense to me. This was a source of large confusion while reading. All claims of “Heisenberg limit” and “gain over classical limits” must be removed or rephrased as classical amplification effects.

Reply: We thank the Referee for the insightful comments regarding our use of the term “Heisenberg limit.” We acknowledge that the Heisenberg limit is most commonly discussed in the context of multi-qubit systems, where it refers to the ultimate precision scaling of $1/N$ with respect to the number of particles or probes. However, as pointed out in several foundational works, the concept of the Heisenberg limit can be generalized beyond the number of qubits to other resources. For example, when the total interaction time is treated as the primary resource, the achievable precision may scale as $1/T$ [Zhou, Sisi, et al. *Nat. Commun.* **9**, 78 (2018); Rafał Demkowicz-Dobrzański, et al. *Phys. Rev. X* **7**, 041009 (2017)]. Another widely adopted resource, especially in sequential strategies, is the number of repeated operations applied to a single probe, which has been used in works such as [Vittorio Giovannetti, et al. *Phys. Rev. Lett.* **96**, 010401

(2006); Marcin Zwierz, et al. *Phys. Rev. Lett.* **105**, 180402 (2010); Rafal Demkowicz-Dobrzanski, et al. *Phys. Rev. Lett.* **113**, 250801 (2014)].

In our manuscript, the section “sensing of remote vector fields” compares our controlled-enhanced sequential scheme with the individual measurement strategy [Hou, Zhibo, et al. *Sci. Adv.* **7**, eabd2986 (2021)]. In this individual strategy, the measurement is performed immediately after one operator acts on the system. The procedure is then repeated $N/3$ times for estimating one parameter— N operators total for estimating all three parameters. The resulting precisions are given by:

$$\delta B_{\text{est}} \geq \frac{\sqrt{3}}{2\sqrt{NT}}, \quad \delta\theta_{\text{est}} \geq \frac{\sqrt{3}}{2\sqrt{N}|\sin(BT)|}, \quad \delta\phi_{\text{est}} \geq \frac{\sqrt{3}}{2\sqrt{N}|\sin(BT)\sin\theta|},$$

which are taken as the standard quantum limit.

When the N operators act sequentially on the system qubit with the total evolution given by $[U_s(\hat{x})]^N = e^{-iN\vec{B}\cdot\vec{\sigma}T}$, here $\hat{x} = (B, \theta, \phi)$ and $\vec{B} = (B \sin \theta \cos \phi, B \sin \theta \sin \phi, B \cos \theta)$. For the estimation of B , this indeed results in an effective amplification $B \rightarrow NB$, which improves the precision of estimating B as $\delta B_{\text{est}} \geq \frac{1}{2NT}$, as correctly pointed out by the referee. The precision for the estimation of θ and ϕ , $\delta\theta_{\text{est}} \geq \frac{1}{2|\sin(NBT)|}$, $\delta\phi_{\text{est}} \geq \frac{1}{2|\sin(NBT)\sin\theta|}$, however, does not necessarily improve in the sequential scheme **without control**.

As a contrast, under the optimal controlled sequential scheme, all three estimators achieve a precision scaling of $1/N$,

$$\delta B_{\text{est}} \geq \frac{1}{2NT}, \quad \delta\theta_{\text{est}} \geq \frac{1}{2N|\sin(BT)|}, \quad \delta\phi_{\text{est}} \geq \frac{1}{2N|\sin(BT)\sin\theta|}.$$

This underscores the importance of control in sequential schemes.

Changes to the manuscript: To avoid potential confusion, we have replaced all claims to the “Heisenberg limit” in the revised main text and SI with the more precise expression such as “ $1/N$ scaling limit”, “*an inverse-linear scaling with sensing circuit depth.*”

C3: The third major criticism is about the maximum likelihood analysis. I do not understand it from the description in the methods section. After acquiring a set of measurement data x_1, \dots, x_n for the given parameters y , one constructs the likelihood function $L(x_1, \dots, x_n|y) = P(x_1|y) * \dots * P(x_n|y)$ and maximizes it $y_{\text{est}}(x_1, \dots, x_n) = \text{argmax}_y L(x_1, \dots, x_n|y)$, providing the maximum

likelihood estimate. This is different from the description in the manuscript. It seems that the authors do not consider actual measurement results and maximize a sort of statistical average of the likelihood function: this is different from taking the statistical average of the maximum likelihood. In other words, “the 2D landscape” of the likelihood must depend on the acquired measurement results. The unusual notion of maximum likelihood extends to the description of the control-enhanced protocol. The methods section must be rewritten to describe the true ML procedure and how it enters the adaptive protocol.

Reply: We thank the Referee for the thoughtful and constructive comments regarding our use of maximum likelihood estimation (MLE). We recognize that the distinction between the actual parameter estimation and the benchmarking procedure may not have been sufficiently clear in the original Methods section, and we have revised the text accordingly.

Our experiment involves two distinct uses of likelihood functions, which serve complementary purposes : (1) to estimate the unknown signal parameters using standard MLE based on experimental data, and (2) to benchmark and visualize the performance of the sensing protocol. These two uses are conceptually and operationally separate, and we now make this distinction explicit in the revised text.

First, **we employ standard maximum likelihood estimation based on experimentally acquired outcomes to extract the unknown signal parameters**. For a fixed control setting \hat{x}_c , we run the sensing circuit n times and collect measurement outcomes $\{y_1, \dots, y_n\}$, where each $y_j \in \{00, 01, 10, 11\}$ for a 2-qubit measurement. The likelihood function is defined as

$$\mathcal{L}(y_1, \dots, y_n | \hat{x}, \hat{x}_c) = \prod_{j=1}^n P(y_j | \hat{x}, \hat{x}_c)$$

where $P(y_j | \hat{x}, \hat{x}_c)$ denotes the probability of observing outcome y_j under signal parameter \hat{x} and a given control \hat{x}_c .

In experiment, under n times of measurement, we count how many times each occurred, say each measurement result $i \in \{00, 01, 10, 11\}$ occur n_i times, where $n = \sum_i n_i$. We denote the experiment measured probability as $P_i^{\text{exp}} = n_i/n$, and the ideal model-predicted probability as

$P_i^{\text{ideal}}(\hat{x}, \hat{x}_c)$. We group identical measurement outcomes and rewrite the likelihood function as

$$\mathcal{L}(y_1, \dots, y_n | \hat{x}, \hat{x}_c) = \prod_{j=1}^n P(y_j | \hat{x}, \hat{x}_c) = \prod_{i \in \{00, 01, 10, 11\}} [P_i^{\text{ideal}}(\hat{x}, \hat{x}_c)]^{n_i}.$$

In actual parameter estimation, we first obtain a set of experimentally measured outcome probabilities P_i^{exp} under a fixed but unknown true signal parameter \hat{x}_0 and fixed control parameter \hat{x}_c . We then define the normalized log-likelihood function

$$\mathcal{L}(\hat{x}, \hat{x}_c) = \sum_i P_i^{\text{exp}} \ln P_i^{\text{ideal}}(\hat{x}, \hat{x}_c), \quad (\text{C3.1})$$

where $P_i^{\text{ideal}}(\hat{x}, \hat{x}_c)$ are the model-predicted probabilities for parameter \hat{x} . The maximum likelihood estimator is given by

$$\hat{x}_{\text{est}} = \arg \max_{\hat{x}} \mathcal{L}(\hat{x}, \hat{x}_c).$$

This approach follows the standard maximum likelihood estimation procedure, where the experimental data implicitly depend on the true parameter \hat{x} , and the estimator \hat{x}_{est} identifies the parameter value that makes the observed data most probable under the model.

Because the optimal control parameters \hat{x}_c depends on true signal parameters and are initially unknown, we implement an adaptive protocol that updates control parameters iteratively. In each round, the signal parameters are estimated via MLE based on measurement outcomes, and the resulting estimator is used to update the control. After K rounds, the data collected under different control settings $\{\hat{x}_c^{(m)}\}$ are combined into a joint likelihood function

$$\mathcal{L}_{\text{joint}}(\hat{x}) = \prod_{m=1}^{K-1} \sum_i P_i^{\text{exp}(m)} \ln [P_i^{\text{ideal}}(\hat{x}, \hat{x}_c^{(m)})],$$

and the final estimator is obtained via

$$\hat{x}_{\text{est}}^{(K)} = \arg \max_{\hat{x}} \mathcal{L}_{\text{joint}}.$$

This process ensures that the adaptive estimation is grounded in cumulative experimental statistics.

Second, **the likelihood landscape benchmark** is not used for parameter estimation but rather **to visualize the structure of the likelihood function and evaluate the sensing protocol's per-**

formance under fixed control settings. This procedure is entirely data-driven and does not rely on model-based simulations.

We first measure outcome probabilities under a fixed true signal parameter and control parameter \hat{x}_c , which serve as reference probabilities $\{P_i^{\text{exp}}\}$. We then scan over a grid of test parameters \hat{x} . For each scanned value, we perform **actual measurements under the same control setting** \hat{x}_c , yielding probabilities $P_i^{\text{exp}}(\hat{x}, \hat{x}_c)$. Using these experimental probabilities, we construct the empirical likelihood function

$$\mathcal{L}' = \sum_i P_i^{\text{exp}} \ln P_i^{\text{exp}}(\hat{x}, \hat{x}_c), \quad (\text{C3.2})$$

which quantifies the statistical similarity between scanned and reference distributions.

The difference between Eq. C3.1 and Eq. C3.2 is that the model-based ideal probabilities are replaced by experimentally measured ones. **Each point in this landscape is derived from real data, and the resulting likelihood function is solely used for visualization, not for parameter estimation.**

Changes to the manuscript: We have revised the Methods section to clarify the distinction between the two uses of likelihood functions in our work. Specifically, we have reorganized the original content into two separate sections: “*Maximum likelihood estimation for remote field sensing*”, which details the actual parameter estimation procedure based on experimental data and how it enters the adaptive protocol, and “*Experimental likelihood benchmark*”, which uses actual measurement datas to visualize the precision landscape and assess the performance of specific control configuration. This restructuring aims to clarify the distinct roles of the two likelihood analyses and address the Referee’s concern.

C4: Finally, since the authors have recorded probability measurement results, I strongly suggest to compute and discuss the classical Fisher information matrix of their system, see for instance J. Phys. A 53, 023001 (2019) and arXiv:2502.17396 for reviews. The inverse of the Fisher information matrix provides the Cramer-Rao sensitivity in the system.

Reply: We thank the Referee for the insightful suggestion. We fully agree that computing the classical Fisher information matrix (CFIM) is essential for assessing whether the quantum Cramér-Rao bound (QCRB) can be saturated in practice.

As the Referee rightly points out, the inverse of the CFIM sets the achievable precision bound for a specific measurement. In our work, we have indeed calculated the CFIM associated with the

Bell-basis measurements employed in our experiment. As detailed in the original Supplementary Information, the CFIM F_C coincides exactly with the quantum Fisher information matrix (QFIM) F_Q in our protocol, indicating that the QCRB is saturated under the measurement performed.

We acknowledge that this point was not sufficiently emphasized in the original submission. To address this, we have revised both the Supplementary Information and the Methods section to clarify our treatment of the CFIM and its relation to the QFIM.

Changes to the manuscript: We have added more explicit derivations of the CFIM, now included in Supplementary Information Sections I.B and I.C., showing that the CFIM coincides exactly with QFIM in our protocol. Additionally, we have updated the Methods sections “*Ultimate precision for remote vector field sensing*”, “*Precision limit for gradient estimation with non-local entanglement*”, and “*Precision limit for gradient estimation with local entanglement*” to include the discussion of the CFIM and refer to the relevant derivations in the Supplementary Information.

Let me now go in more details through the manuscript:

C5: 1) The abstract does not clearly state the novelty relative to existing work. It must explicitly position this work within multiparameter DQS and highlight concrete advances over atomic/photonics experiments.

Reply: We thank the referee for this helpful suggestion. We have rewritten the abstract to (i) explicitly situate the study within multiparameter distributed quantum sensing and (ii) highlight concrete advances over prior atomic/photonics implementations. The revised abstract now reads:

“Quantum metrology has emerged as a powerful tool for timekeeping, field sensing, and precision measurements in fundamental physics. With the advent of distributed quantum metrology, its capabilities have extended to probing spatially distributed parameters across networked quantum systems. However, scalable implementations of distributed quantum metrology with multiparameter estimation remain limited, particularly due to the challenges of generating high-fidelity non-local entanglement and dealing with incompatibilities in multi-parameter quantum metrology. Here we demonstrate distributed multiparameter quantum metrology on a modular superconducting quantum network with low-loss microwave interconnects, a platform that uniquely combines fast gate operations, adaptive control, and deterministic non-local entanglement. Using a control-enhanced sequential protocol, we estimate all three components of a remote vector field,

achieving up to 6.86 dB improvement in precision over the individual strategy. We further perform direct estimation of vector field gradients along two directions across spatially separated nodes, realizing a 3.44 dB gain over local entanglement strategies. These results establish superconducting quantum networks as a competitive and reconfigurable platform for scalable multiparameter distributed quantum metrology.”

C6: 2) The caption of figure 1 should explain what is actually shown in the figure: instead, it provides very general considerations about the system. These consideration must go in the main text and the caption should be rewritten. As far as I understand, the state transfer (orange arrows) is provided by photons: is that true? What is the different between blue and purple regions? The authors mention “inter-module efficiency approaching 99%”: is this an original achievement of this work? Or, do they have a reference for this? Overall, a more detailed discussion of figure 1, with more details presented in the caption is needed. I think the reader needs to know more details about the transfer of entanglement mediated by light in the main text.

Reply: We thank the Referee for pointing out these important details. In response,

1. We have **rewritten the caption of Fig. 1** to clearly explain the key elements of the schematic, including the role of the qubits, the meaning of local vs non-local entanglement, and the function of microwave-based state transfer. As suggested, we have also incorporated a more systematic explanation in the main text.
2. The state transfer between modules is indeed mediated by microwave photons propagating through aluminum coaxial cables, which serve as multi-mode resonators. Through the co-design of ultra-low-loss interconnects, the on-chip integration of impedance transformers, and the development of optimized quantum state transfer protocols, we have achieved inter-module quantum state transfer fidelities approaching 99%, comparable to the performance typically observed within single-chip systems.

We would like to clarify that the 99% inter-module quantum transfer fidelity was first introduced in our previous work [Ref. 56, Niu et al. *Nat. Electron.* **6**, 267 (2023)], which established the viability of our chip-to-chip interconnect technology. That work was subsequently highlighted in Nature [*Nature* **615**, 10 (2023)] as a significant step toward scalable quantum network. While our current system inherits this high-performance infrastructure and benefits significantly from it, the present work is focused on a different frontier—namely, the

application of such a distributed quantum architecture to quantum sensing, particularly the design and experimental validation of our sensing strategy. For this reason, we previously refrained from elaborating in detail on the underlying interconnect technology and its performance.

We fully agree that this aspect is foundational to the viability and success of our distributed sensing approach. To enhance the manuscript’s accessibility, we have now **expanded the discussion in the main text** to better explain the photon-mediated entanglement transfer mechanism and have added appropriate references to our earlier work for completeness.

3. We have further **revised the Experimental setup section** to clarify the hardware-level entanglement distribution scheme and to provide a more detailed account of the physical mechanisms underlying this implementation.

We hope these updates provide a more informative and self-contained presentation of our work.

C7: 3) As far as I understand, the central module A in Fig. 1 is used to generate an entangled state, while the lateral modules are used for sensing. If that is true, why does the mudule A include sensor qubits (dark blue dots)? I am confused. What is the role of ancilla qubits? Please explain.

Reply: We thank the Referee for the thoughtful comment and insightful questions regarding the roles of sensor and ancilla qubits. The Referee is correct that the central module (Module \mathcal{A}) is responsible for generating entangled states. In addition to entanglement generation, Module A also performs the projective measurement on the sensor and ancilla qubits, which is a critical step in extracting the encoded parameters. We acknowledge that this dual role of Module \mathcal{A} may not have been sufficiently emphasized and have revised the manuscript to clarify it.

In our scheme “Sensing of remote vector fields”, each entangled pair is generalized in Module \mathcal{A} . The state of one qubit—the **sensor qubit**—is sent to the sensor modules (Modules \mathcal{B} or \mathcal{C}), where it interacts with the external vector field and accumulates signal-dependent evolution. The other qubit—the **ancilla qubit**—remains in Module \mathcal{A} and does not undergo signal encoding. After the signal encoding, the state of the sensor qubit is transferred back to the central module \mathcal{A} , where a projective measurement on the Bell states is performed together with the ancilla qubit.

The presence of **sensor qubits** in Module \mathcal{A} , as shown in Fig. 1, reflects this round-trip trajectory: although sensing occurs in the lateral modules, both preparation and final measurement take

place in the central module.

Regarding the role of the **ancilla qubit**: although it does not directly interact with the signal, it provides additional degrees of freedom for optimizing probe states, which can be optimized to give a maximal quantum Fisher information matrix (see Supplementary Information Sec.I for derivation). This ancilla-assisted strategy is particularly valuable in multi-parameter estimation, where the optimal probe state often depends on the specific parameters to be estimated. Such parameter dependence can lead to incompatibility among optimal probes, making ancilla-assisted designs crucial for achieving high-precision estimation across multiple parameters.

Changes to the manuscript: We have updated the caption of Fig.1 and accompanying text to better clarify these roles and the flow of qubits across modules. The revised figure caption now reads: *“...The inclusion of both sensor and ancilla qubits reflects their complementary roles in enabling tailored entangled state preparation, signal encoding, and joint measurement for multi-parameter sensing.”*.

C8: 4) As far as I understand the first experiment involves modules A and B, while the second experiment A, B and C: this should be clarified in the last part of the “Experimental Setup” paragraph.

Reply: We thank the Referee for the helpful suggestion. The Referee is correct that the first experiment involves Modules \mathcal{A} and \mathcal{B} , while the second involves Modules \mathcal{A} , \mathcal{B} , and \mathcal{C} . To avoid ambiguity, we have explicitly stated this distinction in the final part of the “Experimental Setup” paragraph.

Changes to the manuscript: The revised sentences in “Experimental setup” paragraph now read: *“In the first experiment, we employ a maximally entangled state between the central module \mathcal{A} and a single sensor module \mathcal{B} to estimate a remote vector field.”* and *“The second experiment uses two sensor modules, \mathcal{B} and \mathcal{C} , to estimate multiple spatial gradients of vector fields. The protocol begins by locally preparing a two-qubit entangled state on the central module \mathcal{A} , followed by coherent transferring the state of each qubit to two different sensor modules.”*

C9: 5) Figure 2 is somehow detached from its description in main text. To me, Fig.2a should show only module A and B, with two qubits, as discussed in the main text. This would be consistent with the other panels, which refer to the estimation of a single vectorial magnetic field, and to the discussion in the main text. At present Fig.2a is confusing. I have strong doubts about the

validity of the maximum likelihood analysis shown in panel d, see discussion above.

Reply: We thank the Referee for the helpful comments regarding Figure 2. We agree that the inclusion of three modules (\mathcal{A} , \mathcal{B} , and \mathcal{C}) in the original schematic of Fig.2a may lead to confusion, particularly since the main text focuses on a single-vectorial-field estimation experiment involving only Modules \mathcal{A} and \mathcal{B} . To address this, we have updated Fig.2a in the main text to display only Modules \mathcal{A} and \mathcal{B} , thereby making it fully consistent with both the experimental configuration and the discussion in the main text.

The original schematic in Fig.2a was intended to illustrate an alternative layout of the gradient estimation protocol employed in a subsequent experiment, where entangled resources are distributed from Module \mathcal{A} to both Modules \mathcal{B} and \mathcal{C} . In that setting, local magnetic fields at Modules \mathcal{B} and \mathcal{C} are independently estimated using a sensor–ancilla network, thereby calculating the gradients. We would like to clarify that this protocol and its associated precision analysis were discussed in detail in the Supplementary Information.

Changes to the manuscript: To enhance the clarity and logical flow between the main text and the Supplementary Information, and to provide a more coherent account of the experimental protocol, we have moved the original schematic—depicting all three modules—to the Supplementary Information (see Section I.C.3 and Fig.S2). In the main text, we have revised Fig.2a and the accompanying text to be fully consistent with both the experimental configuration and discussion.

C10: 6) In the inset of Fig. 2d, the authors show the “landscape of the likelihood function”. How do the author do the maximum likelihood analysis if the likelihood function, as shown in the plot, has clearly a degenerate maxima? I think the reader needs more details about the maximum likelihood analysis and the adaptive protocol in the main text.

Reply: We thank the Referee for this helpful comment. As clarified in our response to the third major point (comment C3), the 2D likelihood landscapes shown in the inset of Fig.2d are part of a benchmark procedure performed under fixed control parameters, and are not used for parameter estimation. Rather, they demonstrate that maximum likelihood estimation can successfully guide the estimate toward the true signal parameters when the control has been experimentally optimized. These landscapes are based on experimental data and serve to illustrate how increasing the number of sequential operations (e.g., from $N = 4$ to $N = 8$) leads to a more localized likelihood

profile (with a degenerate maxima, as the Referee mentioned), indicating enhanced precision. We have revised Method section titled “*Experimental likelihood benchmark*”, which includes the discussion about the 2D landscape in Fig.2d and Fig.3b.

In contrast, the actual parameter estimation is carried out using a standard maximum likelihood estimation (MLE) procedure. As described in the revised Method section titled “*Maximum likelihood estimation for remote field sensing*”, the optimal control parameters are found through an adaptive protocol that iteratively update the control parameters at each round based on the full dataset accumulated up to that point. This iterative refinement avoids ambiguities associated with locally degenerate likelihood features and ensures reliable convergence to the true parameters, as shown in Extended Data Fig. 1.

Changes to the manuscript: To avoid confusion, we have revised Method section titled “*Maximum likelihood estimation for remote field sensing*” and “*Experimental likelihood benchmark*” to give more details and clarify the distinction between the two uses of likelihood functions in our work.

C11: 7) According to my discussion above, all claims about Heisenberg limit of xdB gain should be removed everywhere in the main text and abstract.

Reply: We thank the Referee for pointing this out. To avoid any potential ambiguity or misinterpretation associated with the term “Heisenberg limit,” we have carefully revised the manuscript. Specifically, all instances of “*Heisenberg limit*” have been replaced with more precise and operationally meaningful descriptions, such as “*1/N scaling limits*” or “*an inverse-linear scaling with sensing circuit depth*”, which more accurately reflect the behavior of our protocol. Corresponding revisions have been made in the abstract, main text, and figure captions to ensure consistency and clarity.

C12: 8) Figure 3 presents some problems as Fig. 2. Specifically, the multiple maxima of the likelihood function, the maximum likelihood analysis, as well as the claim of HL. I suggest to replace HL with $1/N$ scaling.

Reply: We thank the Referee for the helpful comments. Similar to Fig. 2, the 2D likelihood landscapes shown in Fig.3 are used solely for benchmarking purposes under fixed control settings and are not employed for actual parameter estimation. We have clarified this point more explicitly

in the revised manuscript to avoid potential confusion.

Furthermore, in line with the Referee’s suggestion, we have replaced all mentions of the “*Heisenberg limit*” in the main text and figure captions with the more accurate description of “*1/N scaling limits*”, “*an inverse-linear scaling with sensing circuit depth.*”

C13: 9) I have appreciated the comparison between local and nonlocal entanglement in Fig.4. I think this is one of the main messages of this manuscript, and real core of distributed sensing. The authors here need to acknowledge Ref. [30] and Phys. Rev. Lett. 121, 130503 (2018) that have provided a similar discussion.

Reply: We thank the Referee for highlighting the importance of the comparison between local and nonlocal entanglement in distributed quantum sensing. We agree that this comparison is a central aspect of our work. We also appreciate the Referee’s observation regarding the omission of relevant prior studies and welcome the opportunity to address this oversight.

Accordingly, we have revised the discussion associated with Fig.4 to include citations to *Nat. Phys.* **15**, 137–142 (2020) (originally Ref.[30]) and *Phys. Rev. Lett.* **121**, 130503 (2018), thereby acknowledging important earlier contributions and strengthening the relevance of this comparison. Specifically, we have added: “*We note that similar comparisons between local and nonlocal strategies have also been made in prior works^{24,68}.*”

C14: 10) Some overselling claims are not necessary and should be avoided. For instance, I can see the present vector field sensing applied to magnetic field sensing, but implications in “geology and astronomy” in order for “enhancing our understanding of celestial bodies and geological structures” is far from obvious and should be avoided, unless precisely explained. A discussion of time and energy scales relevant for superconducting qubits versus atomic/photonic DQS experiment is also needed (see comment above).

Reply: We appreciate the Referee’s feedback and have revised the introduction and conclusion to avoid overly broad application claims. We now focus on concrete and realistic scenarios where superconducting quantum sensors can offer clear advantages, such as fast microwave field sensing and modular magnetometry. Additionally, we have added a discussion in the introduction regarding the nanosecond-scale operation times of superconducting qubits and their implications for probing fast signals—highlighting a key distinction from atomic and photonic DQS platforms.

C15: 11) Not all the references cited in this work are relevant for the present context. At the same time, several relevant works are not cited. I think the first part of the introduction, up to “In this study ...” should be improved. The text (and the references) is very ambiguous regarding the distinction between single- and multi-parameter quantum metrology. The framework of this manuscript is the multiparameter case. This should be made clear both from the text (I suggest to clarify the multiparameter framework immediately) and from the references (a part a few relevant review on single-parameter case, I suggest to address references discussing the multiparameter case), see below.

Reply: We thank the Referee for this valuable and constructive suggestion. In response, we have revised the first part of the introduction to clearly state that our work is situated within the framework of multiparameter quantum metrology. We have clarified the distinction between single- and multi-parameter sensing and emphasized the unique challenges posed by non-commuting generators in the multiparameter setting. In addition, we have updated and refined the citations to better reflect the relevant literature, including key references on multiparameter quantum metrology and distributed sensing. We believe these changes improve the contextual framing and scholarly rigor of the manuscript.

C16: - The sentence “Quantum metrology, which exploits quantum mechanical phenomena such as superposition and entanglement, seeks to improve measurement precision beyond classical limits.” does not refer to distributed scenario. It seems a general sentence about quantum resources for enhanced metrology: I believe reviews such as Nature photonics 5 (4), 222-229 (2011); Journal of Physics A: Mathematical and Theoretical 47, 424006 (2014); Reviews of Modern Physics 90 (3), 035005 (2018) together with major works on quantum metrology such as Physical review letters 96 (1), 010401 (2006) and Physical review letters 102 (10), 100401 (2009) should be cited instead of Refs. [1]-[7].

Reply: We thank the Referee for the helpful suggestion. In response, we have revised the relevant sentence in the Introduction and updated the citations by replacing the original references with the seminal theoretical works and comprehensive reviews recommended by the Referee, now included as [Refs. 1–5]. These updated references more accurately reflect the foundational and

review literature in the field.

C17: - Regarding distributed quantum metrology [or, more usually “Distributed quantum sensing (DQS)”] certainly Refs. [8,9,10,11,13,15] are relevant, other works are less relevant. For instance Ref. [17] is not relevant as it refers to single-parameter scenario. Alternative papers to the patent [16] by the same authors are available, such as *Physical Review Research* 6 (1), 013246 (2024). I also suggest reviews on multiparameter estimation, such as *Physics Letters A* 384 (12), 126311 (2020), and on DQS, such as arXiv:2502.17396. Other relevant publications regarding DQS are *Nature communications* 11 (1), 3817 (2020), which is similar to Ref. [1] but it has been published before and in a higher impact journal; *Phys. Rev. A* 108, 032621 (2021).

Reply: We thank the Referee for the constructive feedback and insightful citation suggestions. We have removed less relevant works and incorporated more appropriate papers, including the recommended review articles and key publications [Refs. 16, 31, 33, 9, 15], which better capture the scope and progress of the field.

C18: - In the present context of vector field sensing, relevant references are *Physical review letters* 116 (3), 030801 (2016); *Nature Communications* 14, 1021 (2023); *npj Quantum Information* 10, 98 (2024).

Reply: We have further updated the references to include the key works on vector field sensing as recommended by the Referee, namely *Phys. Rev. Lett.* **116**, 030801 (2016) [Ref. 30]; *Nat. Commun.* **14**, 1021 (2023) [Ref. 34]; and *npj Quantum Inf.* **10**, 98 (2024) [Ref. 35].

C19: To summarize, certainly I cannot recommend the publication of the present version of the manuscript in *Nature Communication*. On the positive side, this is the first manuscript discussing distributed sensing with superconducting qubits: possible applications and interest must be clarified. On the negative side, I need to mention the misleading notion of Heisenberg limit and the unclear maximum likelihood analysis. In addition, the writing should be revised and the context of multiparameter sensing should be clarified (with discussion and irrelevant references). I am willing to reconsider an revised version of the manuscript, provided the authors address all my criticism.

Reply: We sincerely thank the Referee for carefully reading our manuscript and providing constructive and detailed feedback. We acknowledge that certain key aspects of our work, particularly regarding motivation, terminology, and methodology, may not have been sufficiently clear in the original submission.

In response, we have substantially revised the manuscript to clarify the scope, improve the presentation, and better situate our work within the broader context of quantum sensing. Specifically, we have

1. We have expanded the Introduction and Discussion sections to clearly articulate the motivation for pursuing distributed quantum sensing with superconducting qubits. This includes comparisons with other leading platforms such as photonic and atomic systems, and a clearer explanation of the potential applications and practical relevance of our approach;
2. To avoid ambiguity, we have carefully removed or replaced all references to “Heisenberg limit” and “quantum advantage” with more precise and appropriate language, reflecting the theoretical bounds and actual performance in our context;
3. We now provide an explicit description of the MLE framework used in our analysis, especially its role in parameter inference and explaining how the likelihood landscape is utilized to benchmark sensor performance;
4. We have revised the reference list to remove irrelevant citations, include more appropriate and recent works, and strengthen the connection to existing literature on multiparameter quantum estimation.

A complete list of changes can be found in the “**Summary of Changes**” section below. We greatly appreciate the Referee’s insightful comments, which have helped us significantly improve our manuscript. We hope the revised version satisfactorily addresses all concerns, and we welcome any further suggestions.

IV. SUMMARY OF CHANGES

Here is a brief summary of the major changes. For any additional minor revisions, kindly refer to the PDF files where the changes have been highlighted and marked.

A. main text

1. We have rewritten the abstract to explicitly position our work within the framework of multiparameter distributed quantum sensing, and to clearly highlight the novelty and advances of our superconducting platform relative to prior experiments (following the suggestion of Referee Comment C5).
2. The first paragraph of the introduction has been substantially revised to reflect progress in photonic and atomic platforms, and to explicitly state the challenges of multiparameter estimation with non-commuting generators (addressing Comments B1, C0, C1, C15). We have also updated the citations to include foundational references on quantum metrology, relevant reviews, and key works in distributed and multiparameter quantum metrology, as well as advances in photonic and atomic platforms (addressing Comments C16–C18).
3. The second paragraph of the introduction has been rewritten to highlight the unique features of superconducting qubits for quantum sensing, including their fast operation timescales, practical sensing applications, and recent advances in quantum networking. (Addresses Comments B2, B4, C0, C14).
4. The final paragraph of the introduction has been revised to provide a clearer review of modular quantum architectures and to explicitly state our experimental achievements. We emphasize the novelty of our distributed multiparameter sensing approach and the use of non-local entanglement. The description of the achieved precision gains has been clarified and made more rigorous. (Following the suggestion of B4, C0.)
5. We have revised the first paragraph of the *Experimental Setup* section to provide additional details about the superconducting architecture, including the microwave interconnections, high-fidelity inter-module operations, and the method for generating non-local entangled states across modules. (Addresses Comments B4, C0 and C5)

6. We have updated the title of Fig.1 and completely rewritten its caption to provide a clearer, more informative, and self-contained explanation of the superconducting sensing platform, including its architecture, state transfer mechanism, and role in enabling distributed sensing. (Addresses Comments C6 and C7)
7. We have revised the first paragraph of the *Experimental Setup* section to provide a clearer and more informative description of the experimental protocol. The updated text explicitly states the objectives of parameter estimation and sensor benchmarking performed in our experiments. (Addresses Comment C8)
8. We have revised the second paragraph in the *Sensing of Remote Vector Fields* section to specify that the signal unitary is realized via calibrated quantum gates in order to perform well-controlled benchmark the sensing protocol. We have also explicitly connected the vector field parameters to practical microwave field sensing in superconducting qubit. (Addresses Comment B2 and B12)
9. In the last two paragraphs of the *Sensing of Remote Vector Fields* section, we have revised the discussion to clearly explain the role of maximum likelihood estimation (MLE) in parameter estimation and the use of the likelihood landscape for benchmarking sensor performance, with a reference to the *Methods* section for detailed explanation. We have also revised discussion of the observed $1/N$ scaling and explicitly highlighted that the achieved precision enhancement stems from the use of non-local entangled states combined with a control-enhanced sequential strategy. (Addresses Comment C10)
10. We have updated the title and revised the caption of Fig.2 for improved clarity and self-contained explanation of the protocol. We have modified the schematic in Fig.2a to remove ambiguity and better represent the experimental procedure. (Addresses Comment B6, B7, C9, C10)
11. In the first to third paragraphs of the section *Distributed sensing of vector field gradients*, we have made revisions to provide a clearer and more coherent explanation of the protocol and avoid ambiguity.
12. We have updated the title of Fig.3 and revised depiction of the Bell measurement in Fig. 3a to remove ambiguity. (Addresses Comment B9, B10)

13. We have corrected the y -label (“Sum of variances”) in both Fig. 3 and Fig. 4. (Addresses Comment B11)
14. In the fourth paragraph of the *Distributed sensing of vector field gradients* section, we have added a citation to the supplementary information for the discussion of the non-zero gradient case, revised the explanation of the observed $1/N$ scaling, and included a discussion on how decoherence and control errors contribute to deviations from the theoretical scaling. (Addresses Comment A2, B8)
15. In the last three paragraphs of the same section, we have revised the description of the comparative experiment using only local entanglement (LE) and refined the definition of the precision gain metric to make the comparison between NLE and LE strategies clearer. (Addresses Comment C13)
16. In the penultimate paragraph of the same section, we updated the Fig. 4 discussion on local vs nonlocal entanglement to acknowledge prior related analyses, adding corresponding citations. (Addresses Comment C13)
17. In the *Summary* section, we have revised the first paragraph to clearly state our experimental achievements, and comment on the fidelity of the non-local entangled state. We also updated the final paragraph to provide a more specific discussion of the implications, highlight relevant practical applications, and outline future outlooks for distributed quantum sensing. (Addresses Comment A3, A4, C14)
18. We have revised the *Methods* section “Experimental platform” to align with the expanded discussion of microwave interconnections in the main text, and refined the description of experimental parameters to avoid ambiguity.
19. We have rewritten and reorganized the *Methods* section to clarify the two distinct uses of likelihood functions. The original sections (“Maximum likelihood estimation and likelihood function benchmark” and “Adaptive control-enhanced sensing protocol”) are now split into “Maximum likelihood estimation with adaptive control-enhanced sensing protocol”—estimation from measured outcome counts in a fixed basis (no state tomography), explicitly showing how MLE enters the adaptive loop—and “Experimental likelihood benchmark for control evaluation”—a data-driven visualization/diagnostic not used for estimation

or feedback. We also made explicit that the MLE is the standard log-likelihood constructed from the recorded measurement data. (Addresses Comment B4, B5, C3)

20. We have revised the *Methods* section “Ultimate precision for remote vector field sensing” to explicitly state the achieved precisions for estimating the three parameters, provide a detailed discussion of the classical Fisher information matrix (CFIM) and quantum Fisher information matrix (QFIM), and clarify the definition and implementation of the individual strategy used for comparison. (Addresses Comment A1, C4)
21. In the *Methods* sections “Precision limit for gradient estimation with non-local entanglement” and “Precision limit for gradient estimation with local entanglement”, we have added descriptions on CFIM for the sensing protocol and cited the relevant discussion in supplementary information. (Addresses Comment C4)
22. To state our achieved precision gain more rigorously, we have: (1) Replaced all mentions of the “Heisenberg limit” with more precise expressions such as “ $1/N$ scaling” and “inverse-linear scaling with sensing circuit depth.” (2) Replaced terms like “classical limit” and “classical strategy” with “individual strategy.” (Addresses Comment C2, C11, C12)

B. Supplementary Information

1. In Section I.B, we have added a detailed description of the “individual strategy” (previously referred to as the “classical strategy”) and included its corresponding precision. (Addresses Comment A1)
2. To demonstrate that the observed $1/N$ scaling is not a result of classical signal amplification, we have also included in Section I.B a comparison with the sequential strategy without control. (Addresses Comment C2)
3. In Section I.C.1, we have added a numerical analysis (Fig. S1) along with additional discussion to clarify the feasibility of the proposed adaptive compensation scheme. (Addresses Comments A2, B8)
4. In Sections I.B and I.C, we have provided additional details on the calculation of the classical Fisher information matrix (CFIM) based on the outcome probabilities for each sensing strategy. (Addresses Comment C4)

5. We have added Fig.S2 to illustrate different gradient sensing strategies, including directly measuring the gradient by NLE strategy, and separately measuring vector fields \vec{B}_1 and \vec{B}_2 , and calculating their difference (RS/LE strategy). (Addresses Comment B7, B10)
6. A new subsection and figure (Sec. II.C, Fig. S5) have been added to the Supplementary Information, detailing the gate set calibration processes (Addresses Comment B4).

I. RESPONSE TO REFEREE #2

B0: I have carefully read the responses of the Authors to the points I have made, as well as the issues raised by the other Referees, and at this stage I am happy to recommend the article for publication in Nature Communications. As detailed below, I am satisfied and very thankful for the elaborate responses to my points, which are now resolved within the manuscript and its supplement. I still have a minor issue that should be resolved; however, as this is a presentation detail relevant to the Methods section, I will not need to inspect the manuscript again, as long as the Editor verifies this to be resolved in the next round.

In particular, the Authors have convinced me that, although parts of their protocol have already been demonstrated using the photonic platform, the current superconducting implementation constitutes a substantial and important step, including the novel gradient protocol. Moreover, even though they actually simulate the sensing protocol with pre-calibrated gates, the estimated parameters could represent physical external quantities. Furthermore, the modularity of the setup is indeed an important novelty that I have underestimated in my previous review.

Reply: We thank the Referee for the careful reassessment and for recommending the manuscript for publication. We are pleased that the revisions have addressed the previously raised concerns and that the Referee recognizes the superconducting implementation—particularly the modular architecture and the gradient-based protocol—as a substantial advance beyond earlier demonstrations. We will address the remaining minor presentation issue in the Methods section in the next revision, as indicated.

B1: I still personally think that other signal-processing techniques should be adapted to the modular superconducting-qubits setting that go beyond the estimator construction based on maximising the likelihood (actually minimising the cross-entropy, see below), in particular, for real-time problems when sensing e.g. time-varying microwave signals. However, I agree with the Authors that these are beyond the scope of the results presented, constituting the next step for future research.

Reply: We appreciate the Referee's constructive suggestion and agree that real-time sensing of time-varying microwave signals in modular superconducting-qubit platforms will ultimately

require signal-processing approaches beyond likelihood- or cross-entropy-based estimators. The present study focuses on static parameter estimation, for which the likelihood landscape provides a clear and experimentally accessible benchmark to validate the proposed control-enhanced sensing protocol. Extending this framework to nonstationary signals would involve dedicated dynamical modeling and real-time feedback, and therefore lies beyond the scope of this study. We view the incorporation of more advanced signal-processing techniques as a natural direction for future work, particularly in the context of larger-scale and real-time superconducting quantum sensing networks.

B2: Nonetheless, I would like the Authors to still improve sections of Methods, which they have already restructured into sections “Maximum likelihood estimation for remote field sensing” and “Experimental likelihood benchmark”. I think there is some misunderstanding in the nomenclature, and this has also confused one of the other Referees, and should be unified before publishing the article. Following a classic book such as Kay S. “Fundamentals of Statistical Signal Processing: Estimation Theory”, the MLE is constructed based only on the “model” probability distribution, i.e. for a given outcome “ i ” one maximises $\log[P_i^{model}(x, x_c)]$ over x to obtain the estimate of x , which in the Author’s case would involve $P_{model} = P_{ideal}$. Now, what the Authors do is slightly different, as they maximise $L(x, x_c) = \sum_i P_i^{empirical} \ln[P_i^{model}(x, x_c)]$, which is more commonly referred to in the machine learning community as the minimisation of the cross-entropy loss, see e.g. (4.107-108) in Bishop, C. M. - “Pattern Recognition and Machine Learning” or Section 6.2.5 in Murphy’s “Probabilistic Machine Learning: An Introduction (MIT 2022)”. In particular, operationally the Authors by varying x minimise the Kullback-Leibler distance between $P_{empirical}$ (experiment) and P_{model} (ideal). In that sense, I encourage Authors to be a more elaborate on the estimator construction, as calling it “standard maximal likelihood estimation” is misleading.

Reply: We thank the Referee for this careful and technically precise comment, which helped us identify and correct a potential ambiguity in the terminology used in the Methods section. **We agree that clarifying the estimator construction and its nomenclature is important, and we have revised the text accordingly.**

In our experiment, for a fixed control setting \mathbf{x}_c , the sensing circuit is executed n times, producing outcomes $\{y_1, \dots, y_n\}$, where in our experiment each $y_j \in \{00, 01, 10, 11\}$. Under our model

distribution $P^{\text{model}} \equiv P^{\text{ideal}}(\cdot | \mathbf{x}, \mathbf{x}_c)$, the likelihood function of the full measurement record is

$$\mathcal{L}(y_1, \dots, y_n | \mathbf{x}, \mathbf{x}_c) = \prod_{j=1}^n P^{\text{ideal}}(y_j | \mathbf{x}, \mathbf{x}_c), \quad (1)$$

and the maximum likelihood estimator is defined as $\mathbf{x}_{\text{est}} = \arg \max_{\mathbf{x}} \mathcal{L}(y_1, \dots, y_n | \mathbf{x}, \mathbf{x}_c)$ (see, e.g., Kay, *Fundamentals of Statistical Signal Processing: Estimation Theory*).

Grouping identical outcomes with counts n_i and empirical frequencies $P_i^{\text{exp}} = n_i/n$ (where i labels discrete measurement outcomes), the likelihood can be written in the standard multinomial form

$$\mathcal{L}(y_1, \dots, y_n | \mathbf{x}, \mathbf{x}_c) = \prod_i \left[P_i^{\text{ideal}}(\mathbf{x}, \mathbf{x}_c) \right]^{n_i}. \quad (2)$$

Accordingly, the objective function used in the manuscript, which we denote by the same symbol $\mathcal{L}(\mathbf{x}, \mathbf{x}_c)$, is explicitly defined as the normalized log-likelihood:

$$\mathcal{L}(\mathbf{x}, \mathbf{x}_c) \equiv \frac{1}{n} \log \mathcal{L}(y_1, \dots, y_n | \mathbf{x}, \mathbf{x}_c) = \sum_i P_i^{\text{exp}} \log P_i^{\text{ideal}}(\mathbf{x}, \mathbf{x}_c). \quad (3)$$

Maximizing $\mathcal{L}(\mathbf{x}, \mathbf{x}_c)$ yields exactly the same estimator as maximizing the likelihood function.

As also emphasized in the machine-learning literature, maximizing the normalized log-likelihood above is equivalent to minimizing $D_{\text{KL}}(P^{\text{exp}} || P^{\text{ideal}})$ up to an additive constant that is independent of \mathbf{x} (see Murphy, *Probabilistic Machine Learning: An Introduction*, Sec. 6.2.5). To avoid misunderstanding, we have polished the wording in the Methods to consistently refer to $\mathcal{L}(\mathbf{x}, \mathbf{x}_c)$ as the normalized log-likelihood objective, and we have added the above explicit definition to unify the terminology throughout.

Changes to the manuscript: To avoid any misunderstanding, we have revised the **Methods** section to make this estimator construction explicit and have added an appropriate standard reference. In addition, the inset of Fig. 2d and Fig. 3c have been updated to report variance rather than standard deviation, and the quantum and classical scaling limits are consistently expressed as $1/N^2$ and $1/N$, respectively, with corresponding revisions throughout the Methods.

II. RESPONSE TO REFEREE #3

C0: First of all, I really appreciated the new version of the manuscript: there is a huge improvement in clarity and focus compared with the first version that I reviewed months ago. The manuscript is now well focused and well written; the results and the underlying physics are very clear. The authors now rightly highlight one of the winning aspects of this work, namely the use of superconducting qubits for multiparameter distributed sensing. This platform seems crucial for transferring—via microwave photons—entanglement between distant nodes. This possibility is quite different from more standard atom/ion/photon platforms, where particles need to be physically moved from one place to another in order to realize a distributed scheme. In my opinion, this aspect fully justifies a high-impact publication.

Reply: We thank the Referee for the positive and encouraging assessment.

I have minor comments:

C1: 1. The authors constantly talk about “non-local entanglement,” which sounds odd to me. Yet the first word in the title is “distributed.” I thus suggest replacing “non-local entanglement” with “distributed entanglement” throughout the manuscript. For instance, the sentence “generating high-fidelity non-local entanglement” in the abstract can be replaced with “generating and distributing entanglement across a quantum network”; “To establish non-local entanglement” on page 2 can be replaced with “To distribute entanglement”; and “leverage non-local entanglement” later in the text with “exploit distributed entanglement.”

Reply: We thank the Referee for pointing out this ambiguity in terminology. We have revised the manuscript to replace “non-local entanglement” with “distributed entanglement” in general descriptions, to better align with the distributed nature of the platform and the paper’s title. The term “non-local entanglement” is retained only where it is necessary to distinguish the sensing protocol from strategies that involve strictly local entanglement within individual modules.

C2: 2. At the beginning of the paragraph “Sensing of remote fields,” the authors write “we simulate a setup.” What do they mean? To my understanding, this should be “we realize a setup.”

Reply: We thank the Referee for pointing this out. We have corrected the wording from “sim-

ulate a setup” to “realize a setup” at the beginning of the sections “Sensing of remote fields” and “Distributed sensing of vector field gradients” to accurately reflect the experimental implementation.

C3: 3. Regarding the references. Ref. [14], despite the title, does not discuss multiparameter estimation: it is out of context and can be removed or cited elsewhere, not together with Refs. 6-16. The list of experimental works with photons, Refs. [23-28], should include Hong et al., *Nature Communications* 12, 5211 (2021) and at least one work from the group of Fabio Sciarrino, e.g. *Advanced Photonics* 5 (1), 016005 (2023). Ref. [13] is a News & Views piece in *Nature Photonics*; the authors should cite the original work by Xia et al., *Nature Photonics* 17, 470-477 (2023), but I do not think either work is related to multiparameter sensing. Ref. [32] can be cited together with Ref. [9] for “characterization of distributed signals [6-16]”.

Reply: We thank the Referee for these helpful suggestions regarding the references. In response, we have removed the originally cited Refs. [13] and [14], which are not directly related to multiparameter sensing in the present context. We have added the recommended experimental photonic works, namely Hong et al., *Nat. Commun.* **12**, 5211 (2021), and Cimini et al., *Adv. Photonics* **5**, 016005 (2023). Additionally, we have revised the placement of Ref. [32], citing it together with Ref. [9] under “characterization of distributed signals [6–16]” for improved accuracy and clarity.

To conclude, I congratulate the authors on the impressive work and the revision of the manuscript. Without hesitation, I recommend publication in *Nature Communications*.